# Unified Reinforcement and Imitation Learning for Vision-Language Models

**Byung-Kwan Lee**†
NVIDIA, KAIST
byungkwanl@nvidia.com

**Ryo Hachiuma**
NVIDIA
rhachiuma@nvidia.com

**Yong Man Ro**
KAIST
ymro@kaist.ac.kr

**Yu-Chiang Frank Wang**
NVIDIA, National Taiwan University
frankwang@nvidia.com

**Yueh-Hua Wu**
NVIDIA
krisw@nvidia.com

## Abstract

Vision-Language Models (VLMs) have achieved remarkable progress, yet their large scale often renders them impractical for resource-constrained environments. This paper introduces Unified **R**einforcement and **I**mitation **L**earning (RIL), a novel and efficient training algorithm designed to create powerful, lightweight VLMs. RIL distinctively combines the strengths of reinforcement learning with adversarial imitation learning. This enables smaller student VLMs not only to mimic the sophisticated text generation of large teacher models but also to systematically improve their generative capabilities through reinforcement signals. Key to our imitation framework is an LLM-based discriminator that adeptly distinguishes between student and teacher outputs, complemented by guidance from multiple large teacher VLMs to ensure diverse learning. This unified learning strategy, leveraging both reinforcement and imitation, empowers student models to achieve significant performance gains, making them competitive with leading closed-source VLMs. Extensive experiments on diverse vision-language benchmarks demonstrate that RIL significantly narrows the performance gap with state-of-the-art open- and closed-source VLMs and, in several instances, surpasses them.

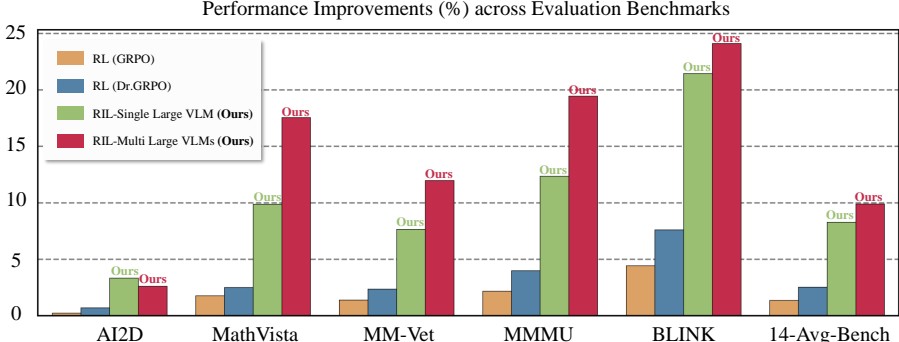

Figure 1: Showing the performance improvements (%) of Qwen2.5-VL-7B [1] across vision-language evaluation benchmarks for AI2D [2], MathVista [3], MM-Vet [4], MMMU [5], BLINK [6], and the average scores for 14 evaluation benchmarks used in Table 1. Note that, we conduct RL on GRPO [7] and advanced GRPO, Dr.GRPO [8], with only answer rewards from LLM-as-a-Judge [9] (see Algorithm 1), and we present RIL based on similarity rewards from single or multi large teacher VLMs and simultaneously answer rewards (see Algorithm 2).

---

† Work Done during Internship.

39th Conference on Neural Information Processing Systems (NeurIPS 2025).

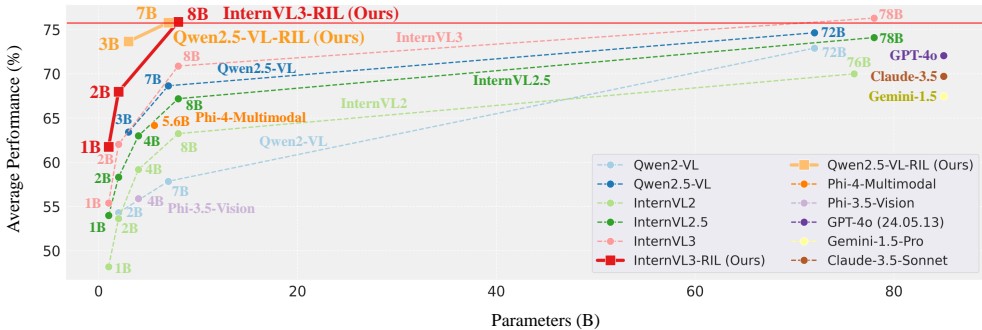

Figure 2: Comparing **RIL**-applied VLMs based on multi large VLMs with diverse open- and closed-source VLMs, under average performance of numerous vision-language evaluation benchmarks: AI2D [2], ChartQA [19], MathVista [3], MMB [20], MM-Vet [4], MMMU [5], MMMU-Pro [21], MMStar [22], BLINK [6], SEED [23], SEED2+ [24], and RealWorldQA (RWQA).

## 1 Introduction

The pursuit of artificial general intelligence (AGI) has gained momentum, with much of the initial progress driven by instruction-tuned large language models (LLMs) [10–15]. Vision-Language Models (VLMs) [16, 1, 17, 18] represent a significant step forward, extending the LLM framework to integrate visual data and thereby achieve multimodal understanding. Their ability to generate insightful, visually-grounded textual responses that increasingly emulate human-like comprehension has made VLMs a focal point of considerable attention. Recognizing their transformative potential, numerous organizations are heavily investing to push the boundaries of what VLMs can achieve.

Historically, the enhancements in vision-language performance have primarily relied on traditional strategies such as scaling up model sizes [15, 25, 26] and expanding visual instruction tuning datasets [27–30, 18]. More recent advancements have explored architectural modifications—for instance, directly altering model structures [31, 32] or integrating auxiliary vision [33, 27, 34] and reasoning modules [35]. Alongside these efforts, the 'think-answer' paradigm [36–42] has gained prominence following DeepSeek-R1's [7] application of reinforcement learning (RL) via GRPO [43]. However, such architectural changes and the verbose 'think' responses preceding answers can significantly increase inference latency and computational memory requirements. These challenges render many powerful VLMs impractical for resource-limited settings like smartphones and augmented reality (AR) devices. As a result, there is a growing imperative within the research community to develop VLM designs that achieve a compelling balance between strong vision-language capabilities, low inference latency, and a lightweight model footprint.

To overcome these challenges, we introduce an approach for developing high-performing, efficient VLMs that avoids architectural modifications and the need for lengthy 'think' responses. Our method, termed Unified Reinforcement and Imitation Learning (RIL), is an efficient training algorithm integrating principles from both GRPO [43] and the GAIL framework [44]. The central objective of RIL is to enable smaller VLMs (e.g., 7B models) to effectively mimic the text generation style of significantly larger VLMs (e.g., 72B models). Prior works [45, 46] have argued that natural language response-based distillation is more effective than high-dimensional feature distillation, emphasizing the importance of utilizing the language head—referred to as the verbalization effect. Motivated by this insight, we began by leveraging the similarity in natural language responses between teacher and student models. Drawing inspiration from how models articulate responses differently, RIL employs an LLM-based discriminator trained to distinguish between the text responses generated by the student (small VLMs) and the teacher (large VLMs). This discriminator's output provides a similarity reward signal, guiding the student VLMs, which act as a generator, to produce responses increasingly akin to those of the teacher. To maintain training stability and prevent the generator or discriminator from overpowering the other, a common issue known as the balance problem [47, 48], the discriminator's architecture initially mirrors that of the generator.

However, the practical implementation of RIL for VLMs entails several specific challenges. Firstly, relying on continuous discriminator scores (ranging from zero to one) to assess similarity to large VLM outputs can introduce ambiguity into the learning signal. To ensure a clearer and more decisive reward, and drawing inspiration from prior works that binarize answer rewards [7, 43], we convert the discriminator's similarity score into a binary value. Secondly, the discriminator, by design, focuses on stylistic similarity and does not inherently verify factual correctness against ground truth

```
SYSTEM: You are a discriminator assistant.
You are provided with Response in <response> </response> tag.
Your task is to determine whether the Response was generated by teacher or student model.
Return 0 if the Response was generated by the teacher model.
Return 1 if the Response was generated by the student model.

USER: <response>{response}</response>
Determine the source of the Response.
Provide your answer as one of the following: 0 (Teacher model) 1 (Student model)
```

Figure 3: Input prompt for Discriminator

answers, which could otherwise lead to performance degradation. We mitigate this by incorporating an LLM-as-a-Judge [9] to provide a separate binary reward signal based on the accuracy of the generated response. A third challenge arises during the GRPO [43] phase: if the student VLMs, due to their more limited knowledge, fail to generate any correct responses for a given task, they struggle to identify effective learning pathways. To provide clearer guidance, especially when the student VLM lacks sufficient domain-specific knowledge compared to larger models, we incorporate text responses from both student and teacher VLMs during this GRPO training step. This strategy not only stabilizes RIL training but also creates opportunities for the student VLM to potentially surpass the performance of larger teachers.

Several key aspects of RIL contribute to its effectiveness. Notably, we observe that sourcing text responses from **multiple large teacher VLMs** significantly boosts student performance beyond what a single teacher can achieve, owing to the richer diversity of responses and the consequently more robust discriminator. This enriched learning environment is further enhanced during GRPO [43] step: by incorporating text responses from both student and teacher VLMs, we provide the student with clearer exemplars for reaching correct answers. RIL also demonstrates particular synergy with distillation-based VLMs; these models show marked performance improvements because their inherent feature alignment through distillation complements RIL's objectives. Beyond these learning dynamics, a crucial practical advantage of RIL is its inference efficiency: trained models do not require an explicit 'think' phase before generating answers, thereby maintaining the same inference speed as the original student VLM.

Furthermore, RIL's reliance on LLM-as-a-Judge [9] for accuracy evaluation offers substantial advantages in applicability. Unlike methods such as DeepSeek-R1 [7] and its variants [39, 49, 41, 37], which often use conventional answer parsing for reward computation and are thus limited to domains with predefined metrics (e.g., math or image grounding), RIL transcends these limitations. For example, conventional parsing struggles with open-ended general visual questions, such as "How to cook this dish?" based on a food image, or "Summarize this movie" from its poster. In addition, the parsing poses a challenge to flexibly compare different types of answers, if a response is "The answer is twenty percent" but its ground truth is "20". The versatile evaluation power of LLM-as-a-Judge [9] enables RIL's successful application to a broad spectrum of visual question answering tasks, fostering strong performance in areas like recognition, OCR, common-sense reasoning, mathematical problem-solving, and chart/document understanding. Finally, because RIL operates purely on generated text responses, it is **agnostic to the specific image embedding strategies or language tokenizers** used by the student or teacher VLMs, ensuring broad compatibility.

To validate our approach, we conduct extensive experiments across diverse vision-language benchmarks. The results compellingly demonstrate that RIL not only significantly narrows the performance gap to state-of-the-art open- and closed-source VLMs but, in several instances, surpasses them. The primary contributions of this work are:

- **Unified Reinforcement and Imitation Learning (RIL)**: We introduce RIL, an efficient and novel training framework for VLMs. It empowers smaller models to emulate the text generation patterns of significantly larger VLMs, leading to substantial enhancements in their overall performance.

- **Broad Applicability and Flexibility**: RIL exhibits wide applicability, functioning effectively with diverse VLMs irrespective of their underlying image embedding strategies or language tokenizers. Furthermore, it preserves the original inference speed by avoiding lengthy intermediate reasoning steps and capably addresses general visual question answering tasks through its integration with LLM-as-a-Judge.

Figure 4: Input prompt for LLM-as-a-Judge [9]

- **Extensive Validation and Significant Performance Gains**: Our comprehensive experimental results consistently show that RIL delivers substantial performance improvements across a variety of vision-language benchmarks. These gains position RIL-trained models as highly competitive with, and sometimes superior to, existing state-of-the-art VLMs.

## 2   Related Works

**Evolution of Vision-Language Models (VLMs).**   Conventional remedies for advancing VLMs have largely centered on scaling up model sizes and expanding datasets to push performance boundaries. Consequently, promising VLMs have emerged, for example, closed-source VLMs: GPT-4o [50], Gemini [51], and Claude-3.5 Sonnet [52], and open-source VLMs: Molmo-72B [53], LLaVA-OneVision-72B [28], NVLM-72B [54], Qwen2.5-VL-72B [1], and InternVL3-78B [18] have followed this paradigm, incorporating large-scale language models such as Qwen2/Qwen2.5-72B [13]. However, the sheer size and computational demands of these models present significant barriers to deployment in resource-constrained environments such as mobile and embedded devices. To address these challenges, CoLLaVO [33] and MoAI [34] utilize computer vision models directly for visual capability, and Mini-Gemini [27], MoVA [55], and Eagle [56] employ multiple vision encoders such as CLIP [57], ConvNeXt [58], DINO-v2 [59], and SAM [60]. In parallel, Meteor [35] explores the efficient way of learning complex reasoning abilities, and TroL [31] and Phantom [32] investigate propagation modification for how we can embed vision-language knowledge as much as possible despite using the same architectures. More recently, since DeepSeek-R1 [7] introduces think-answer process and shows its dramatic performance improvements by using reinforcement learning (RL) such as GRPO [43], many variant models for VLMs has presented LMM-R1 [61], Vision-R1 [37], R1-V [42], OpenVLThinker [62], R1-OneVision [63], R1-Zero [64], and MM-Eureka [65]. They employ think-answer process to VLMs in specific areas such as math problems and object counting tasks. In addition, NoisyRollout [40] injects noise into clean images for distortion and train VLMs to strongly improve visual robustness, thereby understanding visual properties more than before.

**Imitation Learning (IL).**   It originates from the concept of learning expert behavior in robotics. Generative adversarial imitation learning (GAIL) [44] is a foundational framework in this domain. It employs a generator and a discriminator where the generator (e.g., small student VLMs) attempts to replicate expert behavior (e.g., large teacher VLMs) by producing outputs such as actions or trajectories (e.g., text responses) that are indistinguishable from those generated by experts. Meanwhile, the discriminator aims to distinguish between the generator and the expert. By framing imitation learning (IL) as a minimax game, GAIL [44] leverages adversarial training to align the behavior of the generator with that of the expert without explicitly defining a reward function. Instead, GAIL [44] only relies on discriminator scores to evaluate how the generator similarly produces outputs of an expert. This approach has demonstrated strong performance in complex, high-dimensional tasks by directly learning from expert data, making it a widely adopted strategy in IL [66]. Based on its effectiveness, we integrate reinforcement and imitation learning (RIL) for VLMs with four key modifications: (1) combining GRPO [43] and GAIL [44] framework with explicitly reward design, (2) making the output scores of the discriminator binary to stabilize training, (3) incorporating LLM-as-a-Judge [9] to evaluate answer rewards that assess whether the generated text responses from

Table 1: Comparing the performances by using answer rewards from LLM-as-a-Judge [9] with purely RL on GRPO [7] and advanced GRPO [8], and by using similarity and answer rewards from RIL on GAIL [44], under numerous evaluation benchmarks: AI2D [2], ChartQA [19], MathVista [3], MMB/MMBCN [20], MM-Vet [4], MM-Vet-v2 [67], MMMU [5], MMMU-Pro [21], MMStar [22], BLINK [6], SEED [23], SEED2+ [24], and RealWorldQA (RWQA). Note that, for RIL, we set large VLMs as the largest version of same small VLMs, such as Qwen2.5-VL-7B ← Qwen2.5-VL-72B and InternVL3-8B ← InternVL3-78B.

| VLMs | AI2D | ChartQA | MathVista | MMB | MMB$^{CN}$ | MM-Vet | MM-Vet-v2 | MMMU | MMMU-Pro | MMStar | BLINK | SEED | SEED2+ | RWQA |
|---|---|---|---|---|---|---|---|---|---|---|---|---|---|---|
| Qwen2.5-VL-7B | 83.9 | 87.3 | 67.8 | 83.5 | 83.4 | 71.8 | 63.7 | 55.0 | 38.3 | 63.9 | 56.4 | 77.0 | 70.4 | 68.5 |
| w. RL (GRPO) | 84.1 | 88.8 | 69.0 | 83.8 | 83.9 | 72.8 | 63.9 | 56.2 | 40.3 | 65.2 | 58.9 | 77.4 | 70.6 | 69.4 |
| w. RL (Dr.GRPO) | 84.5 | 90.0 | 69.5 | 84.3 | 84.4 | 73.5 | 64.2 | 57.2 | 41.8 | 66.3 | 60.7 | 78.0 | 70.9 | 70.3 |
| w. RIL (Dr.GRPO + GAIL) | **86.7** | **95.4** | **74.5** | **86.8** | **87.2** | **77.3** | **66.1** | **61.8** | **48.2** | **71.1** | **68.5** | **80.7** | **73.0** | **74.2** |
| Qwen2.5-VL-3B | 81.6 | 84.0 | 61.2 | 79.1 | 78.1 | 61.8 | **57.6** | 51.2 | 31.6 | 55.9 | 47.6 | 74.0 | 67.6 | 65.4 |
| w. RL (GRPO) | 81.9 | 87.8 | 61.8 | 80.3 | 79.7 | 62.8 | 55.4 | 52.0 | 33.7 | 57.7 | 52.6 | 75.7 | 68.6 | 66.4 |
| w. RL (Dr.GRPO) | 82.4 | 89.7 | 62.5 | 81.8 | 81.2 | 63.8 | 55.9 | 52.5 | 34.3 | 59.1 | 53.7 | 76.5 | 69.2 | 67.0 |
| w. RIL (Dr.GRPO + GAIL) | **83.0** | **95.4** | **65.2** | **84.8** | **84.2** | **67.4** | **57.6** | **53.7** | **36.8** | **61.2** | **55.5** | **78.7** | **70.3** | **67.7** |
| InternVL3-8B | 85.2 | 86.6 | 71.6 | 83.4 | 82.2 | 78.5 | 66.3 | 62.7 | 41.3 | 68.2 | 55.5 | 77.1 | 69.7 | 70.8 |
| w. RL (GRPO) | 85.9 | 89.6 | 72.3 | 84.6 | 84.2 | 78.7 | 66.4 | 63.8 | 42.5 | 70.0 | 57.1 | 77.9 | 70.6 | 71.3 |
| w. RL (Dr.GRPO) | 86.3 | 91.2 | 72.9 | 85.2 | 85.3 | **79.0** | 66.5 | 64.9 | 42.8 | 70.7 | 57.5 | 78.1 | 71.1 | 72.0 |
| w. RIL (Dr.GRPO + GAIL) | **87.4** | **95.5** | **74.1** | **88.7** | **89.3** | 78.4 | 66.5 | **66.8** | **44.8** | **75.4** | **60.1** | **80.6** | **73.0** | **73.3** |
| InternVL3-2B | 78.7 | 80.2 | 57.0 | 81.1 | 78.4 | **62.2** | 53.9 | 48.6 | 24.9 | 60.7 | 47.0 | 75.0 | 64.6 | 64.3 |
| w. RL (GRPO) | 79.1 | 86.9 | 57.8 | 82.4 | 80.0 | 61.8 | 53.6 | 49.6 | 25.7 | 61.5 | 47.3 | 75.8 | 64.9 | 64.6 |
| w. RL (Dr.GRPO) | 79.8 | 90.3 | 58.5 | 83.2 | 81.3 | 62.1 | 53.8 | 50.5 | 26.4 | 62.4 | 47.9 | 76.5 | 65.4 | 65.2 |
| w. RIL (Dr.GRPO + GAIL) | **81.1** | **93.6** | **63.0** | **85.6** | **83.2** | 61.5 | **54.2** | **52.9** | **27.5** | **63.2** | **48.3** | **78.2** | **66.7** | **66.8** |
| InternVL3-1B | 69.4 | 75.3 | 45.8 | 72.6 | 67.9 | **58.7** | 47.5 | 43.4 | 17.5 | 51.5 | 42.9 | 71.2 | 58.2 | 58.2 |
| w. RL (GRPO) | 69.6 | 83.0 | 46.7 | 73.8 | 69.4 | 57.7 | 47.0 | 42.8 | 17.2 | 52.5 | 42.7 | 71.5 | 58.5 | 58.5 |
| w. RL (Dr.GRPO) | 69.9 | 86.7 | 47.6 | 74.9 | 70.5 | 57.5 | 47.0 | 42.8 | 17.3 | 53.4 | 42.9 | 71.9 | 59.4 | 59.4 |
| w. RIL (Dr.GRPO + GAIL) | **72.0** | **93.6** | **51.6** | **79.6** | **75.7** | 56.9 | 47.4 | **43.1** | **18.3** | **57.4** | **44.1** | **74.2** | **61.4** | **61.4** |

student VLMs are correct, and (4) utilizing the generated text responses from teacher VLMs when GRPO [43] updates student VLMs, serving as a direct guidance to reach the correct text responses. These approach not only stabilizes RIL training but also provides a mechanism for student VLMs to potentially exceed the performance of teacher VLMs. Moreover, RIL do not require an explicit think-answer process, thereby maintaining the same inference, and it is independent of any particular image embedding methods or language tokenizers used in the student or the teacher VLMs.

# 3 Unified Reinforcement and Imitation Learning for VLMs

## 3.1 Core Model Components for RIL

Our RIL framework utilizes several key model components. For the student models—which are the VLMs we aim to enhance—we employ recently released architectures such as Qwen2.5-VL (3B and 7B variants) [1] and InternVL3 (1B, 2B, and 8B variants) [18]. Starting from these pre-trained checkpoints, RIL directly updates their parameters with the goal of developing efficient, high-performing VLMs capable of rivaling leading closed-source systems like GPT-4o [50], Gemini [51], and Claude-3.5 Sonnet [52]. For the teacher models, which provide the target behavior, we select powerful large teacher VLMs recognized for their strong performance on benchmarks like VLM Leaderboard [68], specifically Qwen2.5-VL-72B [1] and InternVL3-78B [18]. The student VLMs are trained to mimic the text generation patterns and response styles of these teacher models. Central to the imitation learning aspect is a discriminator, designed to assess the similarity between student-generated responses and those from the teacher VLMs. To promote training stability and mitigate the balance problem common in adversarial setups [47, 48]—where one component might overpower the other—the discriminator's architecture and initial parameters deliberately mirror those of the student VLMs. Lastly, to evaluate the factual correctness of the generated text, we utilize an LLM-as-a-Judge [9], specifically Qwen2.5-32B [13], following the methodology of Zheng et al. [9]. This model is not trained but is used solely to determine whether the predicted response accurately reflects the meaning of the ground truth.

The subsequent sections will first elaborate on the discriminator's architecture and pre-training (Section 3.2), followed by a detailed exposition of the RIL algorithm and its reward design (Section 3.3).

## 3.2 Discriminator Architecture and Pre-training

With the student (small) VLMs serving as the generator in our framework, the effective pre-training of the discriminator is crucial. This initial training phase equips the discriminator ($D_\phi$) with the ability to reliably differentiate between text responses originating from student VLMs versus teacher

Table 2: Showing the effectiveness of employing multiple large VLMs more than a single VLM, under the numerous evaluation benchmarks equally used in Table 1.

| VLMs | AI2D | ChartQA | MathVista | MMB | MMB$^{CN}$ | MM-Vet | MM-Vet-v2 | MMMU | MMMU-Pro | MMStar | BLINK | SEED | SEED2+ | RWQA |
|---|---|---|---|---|---|---|---|---|---|---|---|---|---|---|
| Qwen2.5-VL-7B | 83.9 | 87.3 | 67.8 | 83.5 | 83.4 | 71.8 | 63.7 | 55.0 | 38.3 | 63.9 | 56.4 | 77.0 | 70.4 | 68.5 |
| w. RIL (Qwen2.5-VL-72B) | 86.7 | 95.4 | 74.5 | **86.8** | **87.2** | 77.3 | 66.1 | 61.8 | 48.2 | **71.1** | 68.5 | 80.7 | **73.0** | 74.2 |
| w. RIL (InternVL3-78B) | **86.8** | 95.5 | 74.6 | 86.7 | 87.1 | 75.8 | 66.0 | 60.9 | 47.1 | **71.1** | 68.1 | **80.8** | 72.7 | **75.4** |
| w. RIL (Both) | 86.1 | **95.6** | **79.7** | 86.3 | 86.5 | **80.4** | **71.1** | **65.7** | **48.5** | **71.1** | **70.0** | 80.5 | 72.8 | 72.8 |
| Qwen2.5-VL-3B | 81.6 | 84.0 | 61.2 | 79.1 | 78.1 | 61.8 | 57.6 | 51.2 | 31.6 | 55.9 | 47.6 | 74.0 | 67.6 | 65.4 |
| w. RIL (Qwen2.5-VL-72B) | 83.0 | 95.4 | 65.2 | 84.8 | 84.2 | 67.4 | 57.6 | 53.7 | 36.8 | 61.2 | 55.5 | 78.7 | 70.3 | 67.7 |
| w. RIL (InternVL3-78B) | 83.3 | 95.4 | 65.1 | 85.0 | **84.7** | 66.6 | 56.3 | 54.9 | 37.7 | 61.1 | 55.3 | **78.8** | 70.2 | 68.2 |
| w. RIL (Both) | **83.9** | **95.7** | **71.7** | **85.2** | 84.5 | **74.7** | **62.8** | **60.7** | **43.7** | **65.2** | **60.8** | 78.7 | **71.4** | **73.2** |
| InternVL3-8B | 85.2 | 86.6 | 71.6 | 83.4 | 82.2 | 78.5 | 66.3 | 62.7 | 41.3 | 68.2 | 55.5 | 77.1 | 69.7 | 70.8 |
| w. RIL (Qwen2.5-VL-72B) | **87.5** | 95.2 | 74.3 | 88.8 | **89.8** | 77.6 | 65.9 | 67.3 | 45.7 | 75.2 | 59.1 | 80.4 | 72.9 | 72.5 |
| w. RIL (InternVL3-78B) | 87.4 | **95.5** | 74.1 | **88.7** | 89.3 | 78.4 | 66.5 | 66.8 | 44.8 | **75.4** | 60.1 | **80.6** | 73.0 | 73.3 |
| w. RIL (Both) | 87.3 | 95.3 | **77.8** | 88.1 | 89.6 | **80.1** | **67.6** | **68.6** | **47.8** | 74.8 | **62.7** | 80.5 | **73.8** | **73.7** |
| InternVL3-2B | 78.7 | 80.2 | 57.0 | 81.1 | 78.4 | 62.3 | 53.9 | 48.6 | 24.9 | 60.7 | 47.0 | 75.0 | 64.6 | 64.3 |
| w. RIL (Qwen2.5-VL-72B) | **81.1** | 93.6 | 61.9 | 85.3 | 83.0 | 62.6 | 52.8 | 52.6 | 26.6 | 63.6 | 48.8 | 78.0 | **67.1** | 66.3 |
| w. RIL (InternVL3-78B) | **81.1** | 93.6 | 63.0 | **85.6** | **83.2** | 61.5 | 54.2 | 52.9 | 27.5 | 63.2 | 48.3 | **78.2** | 66.7 | 66.8 |
| w. RIL (Both) | 80.7 | **94.0** | **67.4** | 84.7 | 82.2 | **70.7** | **58.4** | **56.6** | **32.3** | **63.9** | **51.5** | 77.8 | 66.8 | **69.3** |
| InternVL3-1B | 69.4 | 75.3 | 45.8 | 72.6 | 67.9 | 58.7 | 47.5 | 43.4 | 17.5 | 51.5 | 42.9 | 71.2 | 58.2 | 58.2 |
| w. RIL (Qwen2.5-VL-72B) | 72.2 | 93.6 | 50.9 | **79.9** | **75.8** | 56.1 | 47.3 | 43.6 | 18.0 | 58.0 | 43.6 | 74.6 | 60.7 | 61.0 |
| w. RIL (InternVL3-78B) | 72.0 | 93.6 | 51.6 | 79.6 | 75.7 | 56.9 | 47.4 | 43.1 | 18.3 | 57.4 | 44.1 | 74.2 | 61.4 | 61.4 |
| w. RIL (Both) | **73.0** | **94.1** | **55.5** | 79.1 | 75.9 | **62.9** | **50.7** | **49.7** | **19.0** | **60.5** | **46.9** | **74.7** | **61.9** | **63.7** |

(large) VLMs. Without adequate pre-training, $D_\phi$ would yield unreliable scores, undermining its utility and potentially degrading overall model performance. To prepare for pre-training, we first collect a dataset of responses. For each given question $q$, $N$ text responses are generated from both the student VLMs, denoted as $\{\mathbf{o}_i^{(\mathbf{s})}\}_{i=1}^N$, and the teacher VLMs, denoted as $\{\mathbf{o}_i^{(\mathbf{t})}\}_{i=1}^N$. Using this collection, the discriminator $D_\phi$, parameterized by $\phi$, is trained to maximize the following objective:

$$\max_\phi \mathcal{L}(\phi) = \frac{1}{N} \sum_{i=1}^N \left\{ \log D_\phi(q, o_i^{(\mathbf{s})}) + \log(1 - D_\phi(q, o_i^{(\mathbf{t})})) \right\}. \qquad (1)$$

According to this objective, $D_\phi$ learns to output a score approaching 'one value' for responses from student VLMs and 'zero value' for those from teacher VLMs. For practical implementation, given that the discriminator's backbone is language-based (mirroring the student VLMs), the generated responses $o$ are formatted using a specific prompt (detailed in Fig. 3) to ensure proper textual understanding. To elicit the scalar discrimination score, the standard language head of the VLM (typically mapping to vocabulary logits, ($\mathbb{R}^{d \times v}$) is replaced with a linear discriminator head ($\mathbb{R}^{d \times 1}$). The input to this head is the representation of the final sequence token from the last layer of the backbone model, from which the score is directly computed with the sigmoid function.

### 3.3 Mimicking Large Teacher VLMs with Similarity and Answer Rewards

The RIL process commences after an initial supervised finetuning (SFT) phase for the student (small) VLMs $\pi_\theta$ using a comprehensive visual instruction tuning dataset (see Appendix C). This SFT stage acts as a crucial warm-up, acclimating $\pi_\theta$ to the target data distribution. Once the student VLMs and the pre-trained discriminator $D_\phi$ (from Section 3.2) are prepared, the RIL loop begins. For each question $q$ in a training batch, we generate $G$ text responses from the current student VLMs, $\{o_i^{(\mathbf{s})}\}_{i=1}^G$, and retrieve $G$ responses corresponding the question $q$ from the teacher (large) VLMs, $\{o_i^{(\mathbf{t})}\}_{i=1}^G$. We note that pre-generating and caching the responses from teacher VLMs (e.g., during the discriminator's pre-training) can significantly reduce RIL training time. In the end, this yields a combined set of $2G$ responses, $\{o_i\}_{i=1}^{2G}$, for each question. Within each RIL iteration, we first update the discriminator $D_\phi$ using Eq. (1) on these $2G$ responses. This step ensures that $D_\phi$ maintains its ability to distinguish between student and teacher outputs as the student VLMs evolve. Next, the student VLMs $\pi_\phi$ are updated using an objective derived from Dr.GRPO [8], an advanced variant of GRPO that provides unbiased advantage estimates $\{\hat{A}_i\}_{i=1}^{2G}$. The objective function is:

$$\max_\theta \mathcal{L}(\theta) = \frac{1}{2G} \sum_{i=1}^{2G} \left\{ \min \left[ r_i(\theta)\hat{A}_i, \text{clip}\left(r_i(\theta), 1-\epsilon, 1+\epsilon\right)\hat{A}_i \right] - \beta \mathcal{D}_{\text{KL}}\left(\pi_\theta \mid \pi_{\text{ref}}\right) \right\},$$

$$\text{where} \quad r_i(\theta) = \frac{\pi_\theta(o_i \mid q)}{\pi_{\theta_{\text{old}}}(o_i \mid q)}, \quad \hat{A}_i = R(q, o_i) - \frac{1}{2G} \sum_{j=1}^{2G} R(q, o_j). \qquad (2)$$

Here, $r_i(\theta)$ is the probability ratio between the current policy $\pi_\theta$ and the policy before the update $\pi_{\text{old}}$, $\pi_{\text{ref}}$ is typically the initial SFT model, $\epsilon$ is a clipping hyperparameter, and $\beta$ controls the KL-

Table 3: Comparing **RIL**-applied VLMs with standard or smaller model size open-source VLMs.

| VLMs | AI2D | ChartQA | MathVista | MMB | MMB$^{CN}$ | MM-Vet | MMMU | MMMU-Pro | MMStar | BLINK | SEED | SEED2+ | RWQA |
|---|---|---|---|---|---|---|---|---|---|---|---|---|---|
| LLaVA-OneVision-7B [28] | 81.4 | 80.0 | 63.2 | 80.8 | - | 57.5 | 48.8 | 24.1 | 61.9 | 53.0 | 76.7 | 65.4 | 69.9 |
| InternVL2-8B [69] | 83.8 | 83.3 | 58.3 | 81.7 | 81.2 | 54.2 | 49.3 | 29.0 | 61.5 | 50.9 | 75.4 | 67.3 | 64.2 |
| MiniCPM-V2.5-8B [70] | 78.4 | - | 54.3 | 77.2 | 74.2 | 52.8 | 45.8 | - | 51.8 | - | 72.3 | 61.4 | 63.5 |
| MiniCPM-V2.6-8B [70] | 82.1 | - | 60.6 | - | - | 60.0 | 49.8 | 27.2 | 57.5 | 55.2 | 74.0 | 65.7 | 65.0 |
| MiniCPM-o2.6-8B [70] | 86.1 | 86.9 | 73.3 | - | - | 67.2 | 50.9 | - | 63.3 | 53.9 | 75.5 | 67.9 | 68.0 |
| Ovis2-8B [71] | 86.6 | - | 71.8 | - | - | 65.1 | 57.4 | - | 64.6 | 54.3 | 77.2 | 70.4 | 72.5 |
| Ovis2-16B [71] | 86.3 | - | 73.7 | - | - | 68.4 | 60.7 | - | 67.2 | 59.0 | 77.7 | 72.1 | **74.1** |
| Qwen2-VL-7B [16] | 77.5 | 83.0 | 58.2 | 83.0 | 80.5 | 62.0 | 54.1 | 30.5 | 60.7 | 53.8 | 76.0 | 68.6 | 68.5 |
| InternVL2.5-8B [17] | 84.8 | 84.8 | 64.4 | 84.6 | 82.6 | 62.8 | 56.0 | 34.3 | 63.2 | 54.8 | 76.8 | 69.7 | 70.1 |
| Qwen2.5-VL-7B [1] | 83.9 | 87.3 | 67.8 | 83.5 | 83.4 | 71.8 | 55.0 | 38.3 | 63.9 | 56.4 | 77.0 | 70.4 | 68.5 |
| InternVL3-8B [18] | 85.2 | 86.6 | 71.6 | 83.4 | 82.2 | 78.5 | 62.7 | 41.3 | 68.2 | 55.5 | 77.1 | 69.7 | 70.8 |
| Qwen2.5-VL-**RIL**-7B (Both) | 86.1 | **95.6** | **79.7** | 86.3 | 86.5 | **80.4** | 65.7 | **48.5** | 71.1 | **70.0** | **80.5** | 72.8 | 72.8 |
| InternVL3-**RIL**-8B (Both) | **87.3** | 95.3 | 77.8 | **88.1** | **89.6** | 80.1 | **68.6** | 47.8 | **74.8** | 62.7 | 80.5 | **73.8** | 73.7 |
| Phi-3.5-Vision-4B [72] | 77.8 | 81.8 | 43.9 | 76.0 | 66.1 | 43.2 | 43.0 | 19.7 | 47.5 | 58.3 | 69.7 | 62.2 | 53.6 |
| Phi-4-Multimodal-5.6B [73] | 83.0 | 81.4 | 65.8 | **86.7** | - | 51.9 | 56.0 | 38.5 | 58.9 | 42.1 | 73.2 | 68.5 | 64.1 |
| Ovis2-4B [71] | **85.7** | - | 69.6 | - | - | 65.5 | 49.0 | - | 61.9 | 53.0 | 76.2 | 69.3 | 71.1 |
| InternVL2-4B [69] | 78.9 | 81.5 | 58.6 | 78.6 | 73.9 | 51.0 | 34.3 | 32.7 | 53.9 | 46.1 | 73.2 | 63.9 | 60.7 |
| InternVL2.5-4B [69] | 81.4 | 84.0 | 60.5 | 81.1 | 79.3 | 60.6 | 52.3 | 32.7 | 58.7 | 50.8 | 74.8 | 66.9 | 64.3 |
| Qwen2.5-VL-3B [1] | 81.6 | 84.0 | 61.2 | 79.1 | 78.1 | 61.8 | 51.2 | 31.6 | 55.9 | 47.6 | 74.0 | 67.6 | 65.4 |
| Qwen2.5-VL-**RIL**-3B (Both) | 83.9 | **95.7** | **71.7** | 85.2 | 84.5 | 74.7 | **60.7** | 43.7 | 65.2 | 60.8 | 78.7 | 71.4 | 73.2 |
| InternVL2-2B [69] | 74.1 | 76.2 | 46.3 | 73.2 | 70.9 | 39.5 | 34.3 | 18.2 | 49.8 | 43.8 | 70.9 | 60.0 | 57.3 |
| Qwen2-VL-2B [16] | 60.2 | 73.5 | 43.0 | 74.9 | 73.5 | 49.5 | 41.1 | 21.2 | 47.5 | 45.2 | 72.4 | 61.2 | 62.6 |
| Aquila-VL-2B [17] | 75.0 | 76.5 | 59.0 | - | - | 43.8 | 47.4 | - | 54.9 | 34.1 | 73.9 | 63.0 | 65.0 |
| Ovis2-2B [71] | **82.7** | - | 64.1 | - | - | 58.3 | 45.6 | - | 56.7 | 47.9 | 74.4 | **67.4** | 66.0 |
| InternVL2.5-2B [17] | 74.9 | 79.2 | 51.3 | 74.7 | 71.9 | 60.8 | 43.6 | 23.7 | 54.3 | 44.0 | 73.2 | 60.0 | 60.1 |
| InternVL3-2B [18] | 78.7 | 80.2 | 57.0 | 81.1 | 78.4 | 62.2 | 48.6 | 24.9 | 60.7 | 47.0 | 75.0 | 64.6 | 64.3 |
| InternVL3-**RIL**-2B (Both) | 80.7 | **94.0** | 67.4 | 84.7 | 82.2 | 70.7 | **56.6** | 32.3 | 63.9 | 51.5 | 77.8 | 66.8 | 69.3 |
| LLaVA-OneVision-0.5B [28] | 57.1 | 61.4 | 34.8 | 61.6 | 55.5 | 32.2 | 31.4 | - | 37.7 | 52.1 | 66.6 | 45.7 | 55.6 |
| Ovis2-1B [71] | **76.4** | - | **59.4** | - | - | 50.0 | 36.1 | - | 52.1 | 44.0 | 71.7 | 61.4 | **63.9** |
| InternVL2-1B [69] | 64.1 | 72.9 | 37.7 | 65.4 | 60.7 | 32.7 | 36.7 | 14.8 | 45.6 | 38.6 | 65.2 | 54.3 | 50.3 |
| InternVL2.5-1B [17] | 69.3 | 75.9 | 43.2 | 70.7 | 66.3 | 48.8 | 40.9 | 18.8 | 51.3 | 42.0 | 70.4 | 59.0 | 57.5 |
| InternVL3-1B [18] | 69.4 | 75.3 | 45.8 | 72.6 | 67.9 | 58.7 | 43.4 | 17.5 | 51.5 | 42.9 | 71.2 | 58.2 | 58.2 |
| InternVL3-**RIL**-1B (Both) | 73.0 | **94.1** | 55.5 | **79.1** | **75.9** | 62.9 | **49.7** | 19.0 | 60.5 | 46.9 | 74.7 | 61.9 | 63.7 |

divergence penalty. The total reward $R(q, o_i)$ for each response $o_i$ is a crucial composite signal, comprising two binary components:

1. **Similarity Reward**: This indicates if the response $o_i$ resembles those from teacher VLMs, denoted by $\mathbb{1}(D_\phi(q, o_i) < 0.5)$. A discriminator score below 0.5 suggests higher similarity to teacher responses, consistent with $D_\phi$ being trained to output 'zero value' for teacher samples (Eq. 1).

2. **Answer Reward**: It assesses the factual correctness of $o_i$ against a ground truth answer a, determined by LLM-as-a-Judge$(q, a, o_i)$. This evaluation uses the prompt detailed in Fig. 4.

These two rewards are summed to form the overall reward $R(q, o_i)$. The inclusion of the answer reward is essential, as the discriminator alone primarily captures stylistic similarity and does not guarantee factual accuracy, the absence of which could lead to significant performance issues. This dual-reward strategy ensures that mimicking large teacher VLMs involves not just replicating surface-level text patterns but also encompasses the deeper 'verbalization effect' [45], which characterizes how proficient VLMs articulate correct and contextually appropriate answers. The comprehensive RIL procedure is detailed in Algorithm 2.

## 4 Experiments

### 4.1 Dissecting RIL

To understand the distinct contributions of its components, we first analyze RIL by comparing it against reinforcement learning (RL) baselines that utilize only answer rewards. Specifically, we evaluate RL agents trained solely with GRPO [43] or its advanced variant, Dr.GRPO [8], without the discriminator or the associated similarity rewards. As shown in Table 1, RIL, which integrates reinforcement and imitation learning elements based on Dr.GRPO and GAIL [44], demonstrably outperforms these RL-only approaches, confirming the efficacy of our unified training algorithm.

A key finding is the substantial benefit of employing multiple large teacher VLMs. As detailed in Table 2, using responses from a combination of large models (specifically Qwen2.5-VL-72B [1] and InternVL3-78B [18]) yields significantly better vision-language performance than relying on any single teacher. This advantage arises from the richer diversity of textual responses, which strengthens the discriminator's training. Furthermore, incorporating this diverse set of high-quality responses from teacher VLMs directly into the GRPO optimization step provides student VLMs with clearer and

Table 4: Comparing **RIL**-applied VLMs with large size open-source and closed-source VLMs.

| VLMs | AI2D | ChartQA | MathVista | MMB | MM-Vet | MM-Vet-v2 | MMMU | MMMU-Pro | MMStar | BLINK | SEED | SEED2+ | RWQA |
|---|---|---|---|---|---|---|---|---|---|---|---|---|---|
| NVLM-72B [54] | 85.2 | 86.0 | 66.6 | - | 58.9 | - | 59.7 | - | 63.7 | 48.0 | 75.5 | 68.4 | 69.9 |
| LLaVA-OV-72B [28] | 85.6 | 83.7 | 67.5 | 85.8 | 60.6 | - | 56.8 | 31.0 | 65.8 | 55.4 | 77.5 | - | 71.9 |
| Molmo-72B [53] | 83.4 | 87.3 | 58.6 | - | 61.1 | - | 54.1 | - | 63.3 | 49.7 | 74.6 | 67.6 | 73.7 |
| Qwen2-VL-72B [16] | 88.1 | 88.3 | 70.5 | 86.5 | 74.0 | 68.7 | 64.5 | 46.2 | 68.3 | 60.5 | 77.9 | 72.3 | 77.8 |
| Qwen2.5-VL-72B [1] | 88.7 | 89.5 | 74.2 | 88.6 | 76.9 | **76.2** | 68.2 | 46.2 | 70.8 | 64.4 | 79.5 | 73.0 | 75.7 |
| InternVL2-76B [69] | 87.6 | 88.4 | 65.5 | 86.5 | 65.7 | 68.4 | 62.7 | 40.0 | 67.4 | 56.8 | 77.6 | 69.7 | 72.2 |
| InternVL2.5-78B [28] | 89.1 | 88.3 | 72.3 | 88.3 | 72.3 | 65.5 | 70.1 | 48.6 | 69.5 | 63.8 | 77.0 | 71.3 | **78.7** |
| InternVL3-78B [18] | **89.7** | 89.7 | 79.0 | 89.0 | **81.3** | 70.0 | 72.2 | 47.3 | 72.5 | 66.3 | 78.7 | 71.9 | 78.0 |
| Claude-3.5-Sonnet [52] | 81.2 | 90.8 | 67.7 | 82.6 | 70.1 | 71.8 | 68.3 | 51.5 | 65.1 | 60.1 | 61.7 | 71.7 | 65.8 |
| Claude-3.7-Sonnet [52] | 82.5 | - | 66.8 | - | 70.0 | - | 71.0 | - | 65.1 | 56.6 | 74.3 | 67.6 | 55.4 |
| Gemini-1.5-Pro [51] | 79.1 | 87.2 | 63.9 | 73.9 | 64.0 | 66.9 | 62.2 | 46.9 | 59.1 | 59.1 | 76.0 | 70.8 | 67.5 |
| Gemini-2.0-Flash [51] | 83.1 | - | 70.4 | **90.0** | 73.6 | - | 69.9 | **54.4** | 69.4 | 64.0 | - | - | 72.3 |
| GPT-4o (24.05.13) [74] | 84.6 | 85.7 | 63.8 | 83.4 | 69.1 | 71.0 | 69.1 | 51.9 | 64.7 | 68.0 | 77.1 | 72.0 | 75.4 |
| GPT-4.1 (25.04.14) [74] | 85.9 | - | 70.4 | - | 78.8 | - | **74.0** | - | 69.8 | 68.5 | 78.0 | 73.1 | **78.7** |
| Qwen2.5-VL-**RIL**-7B (Both) | 86.1 | **95.6** | **79.7** | 86.3 | 80.4 | 71.1 | 65.7 | 48.5 | 71.1 | **70.0** | 80.5 | 72.8 | 72.8 |
| InternVL3-**RIL**-8B (Both) | 87.3 | 95.3 | 77.8 | 88.1 | 80.1 | 67.6 | 68.6 | 47.8 | **74.8** | 62.7 | 80.5 | **73.8** | 73.7 |

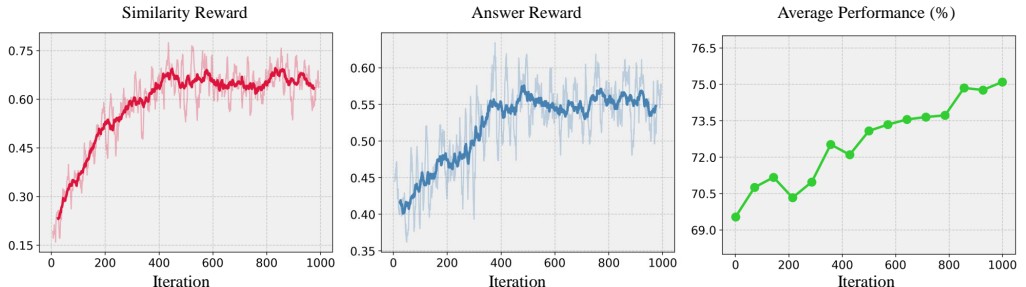

Figure 5: Illustrating training dynamics of small VLMs during RIL, where (Left) showing the evolution of similarity rewards over training iterations and (Mid) accuracy rewards obtained using LLM-as-a-Judge [9], ensuring that generated responses are both contextually appropriate and factually correct. (Right) displaying the overall average performance of evaluation benchmarks used in Table 1.

more varied exemplars of correct answer generation. To be simplified in the manner of an algorithm, given a response 'T1' from one teacher VLM and 'T2' from another teacher VLM, and a response 'S' from the student VLM, then the discriminator is trained to output zero when 'S' is provided and one when either 'T1' or 'T2' is provided. Additionally, the rewards and their advantages for 'S', 'T1', and 'T2' are computed based on similarity and answer quality in order to train the student VLM by using RIL's objective loss.

Beyond these component analyses, our RIL-trained student VLMs exhibit highly competitive performance against a range of recent open-source and closed-source models. Comparisons across various model sizes (detailed in Table 3 for smaller models and Table 4 for larger ones) and diverse vision-language benchmarks (illustrated in Figures 1 and 2) show that RIL not only substantially narrows the performance gap to state-of-the-art VLMs but, in several instances, surpasses them.

Finally, the training dynamics of RIL, depicted in Figure 5, offer further insight. Over the course of training, RIL consistently increases both the similarity rewards (from the discriminator) and the answer rewards (from LLM-as-a-Judge [9]). It indicates that the student VLMs are progressively generating responses that are both more stylistically aligned with teacher models and factually correct, with these improving reward signals directly translating into significant overall performance gains.

For a comprehensive account of the specific training procedures, hyperparameter settings, evaluation setups, and dataset details that ensure the reproducibility of these, please refer to Appendix C. Using 256 NVIDIA A100 GPUs, pre-training the discriminator on 1.2M samples (40K [number of samples] × 16 [generated responses] × 2 [for both teacher and student]) takes approximately 1 to 3 days. The SFT step on the 4M-sample SFT dataset takes around 3 to 5 days. Conducting the RIL loop for the sampled 40K data requires an additional 3 to 5 days using 8 NVIDIA A100 GPUs.

## 4.2 Impacts of Hyperparameters, Techniques, and Distillation

We systematically investigate the sources of RIL's effectiveness by analyzing key training hyperparameters, component design choices, and its interplay with other learning techniques. Note that, for the optimization objective of student VLMs in Eq. (2), the clipping hyperparameter $\epsilon$ is consistently set to 0.2. Our first analyses focus on core optimization settings. We examine the number of update iterations, $\mu$, for the discriminator and student VLMs within each RIL cycle (related to Algorithm 2).

Table 5: Analyzing the impacts of training hyperparameters, techniques for stabilization, and the effect of distillation. Note that, these tables (a)-(c) set a backbone model as Qwen2.5-VL-7B [1] and apply RIL to this model in order to validate it by controlling multiple factors .

(a) Discriminator & small VLMs Iteration ($\mu$)

| $\mu$ | AI2D | MathVista | MMB | MM-Vet | MMMU | BLINK |
|---|---|---|---|---|---|---|
| 1 | **86.1** | **79.7** | **86.3** | **80.4** | **65.7** | **70.0** |
| 2 | 86.1 | 79.0 | 86.3 | 80.4 | 65.3 | 69.2 |
| 4 | 85.5 | 78.8 | 85.5 | 79.5 | 65.0 | 68.4 |
| 6 | 85.1 | 77.5 | 85.2 | 78.0 | 63.8 | 68.0 |
| 8 | 84.5 | 76.2 | 84.7 | 77.4 | 62.5 | 66.7 |
| 10 | 84.3 | 75.0 | 84.5 | 77.2 | 61.9 | 65.5 |
| 12 | 84.0 | 73.8 | 84.4 | 76.9 | 61.0 | 64.2 |

(b) KL-divergence Penalty Coefficient ($\beta$)

| $\beta$ | AI2D | MathVista | MMB | MM-Vet | MMMU | BLINK |
|---|---|---|---|---|---|---|
| 0.00 | 77.1 | 59.9 | 73.2 | 60.2 | 42.9 | 45.0 |
| 0.01 | 86.1 | 79.7 | 86.3 | 80.4 | 65.7 | 70.0 |
| 0.05 | 85.7 | 77.3 | 85.7 | 78.5 | 63.5 | 68.3 |
| 0.10 | 85.2 | 74.9 | 85.2 | 76.6 | 63.0 | 65.8 |
| 0.20 | 84.8 | 74.4 | 84.5 | 75.8 | 62.5 | 64.2 |
| 0.50 | 84.3 | 74.0 | 84.8 | 75.9 | 61.3 | 62.0 |
| 1.00 | 84.2 | 73.9 | 84.3 | 74.8 | 60.4 | 61.7 |

(c) Importance of VLM Modules

| Module | MathVista | MMB | MM-Vet | MMMU |
|---|---|---|---|---|
| Full Training | **79.7** | **86.3** | **80.4** | **65.7** |
| w.o. Word-Embed | 75.4 | 85.5 | 75.1 | 62.2 |
| w.o. Self-Attn | 73.7 | 84.3 | 75.2 | 61.3 |
| w.o. FFN-Gate | 76.5 | 86.0 | 78.8 | 63.5 |
| w.o. FFN-Up | 77.3 | 86.0 | 79.0 | 64.1 |
| w.o. FFN-Down | 78.4 | 86.1 | 79.6 | 64.8 |
| w.o. Layer-Norm | 79.7 | 86.3 | 80.4 | 65.7 |
| w.o. Lang-Head | 74.9 | 85.2 | 73.9 | 62.0 |

(d) Importance of discriminator $D_\phi$

| VLMs | $D_\phi$ | MathVista | MMB | MM-Vet | MMMU |
|---|---|---|---|---|---|
| Qwen2.5-VL-7B | ✗ | 69.5 | 84.3 | 73.5 | 57.2 |
| | △ | 75.3 | 85.5 | 75.1 | 61.2 |
| | ✓ | **79.7** | **86.3** | **80.4** | **65.7** |
| InternVL3-8B | ✗ | 72.9 | 85.2 | 79.0 | 64.9 |
| | △ | 75.3 | 86.6 | 79.5 | 66.5 |
| | ✓ | **77.8** | **88.1** | **80.1** | **68.6** |

(e) Effect of Answer Rewards (AR) and SFT

| VLMs | AR | MathVista | MMB | MM-Vet | MMMU |
|---|---|---|---|---|---|
| Qwen2.5-VL-7B | ✗ | 65.2 | 82.1 | 69.9 | 53.6 |
| | △ | 65.3 | 82.5 | 70.3 | 53.8 |
| | ✓ | **79.7** | **86.3** | **80.4** | **65.7** |
| | ✓w.o. SFT | 73.5 | 84.1 | 76.4 | 60.7 |
| InternVL3-8B | ✗ | 69.3 | 81.4 | 75.4 | 59.3 |
| | △ | 70.0 | 82.1 | 76.2 | 60.2 |
| | ✓ | **77.8** | **88.1** | **80.1** | **68.6** |
| | ✓w.o. SFT | 75.2 | 85.3 | 79.4 | 64.9 |

(f) Effect of Distillation

| VLMs | RIL | MathVista | MMB | MM-Vet | MMMU |
|---|---|---|---|---|---|
| MiniLLM-7B | ✗ | 61.3 | 84.4 | 65.1 | 57.4 |
| | ✓ | **70.8** | **85.3** | **73.3** | **66.1** |
| DistilLLM-7B | ✗ | 61.5 | 84.8 | 65.3 | 57.9 |
| | ✓ | **71.0** | **85.4** | **73.9** | **67.2** |
| LLaVA-KD-7B | ✗ | 61.8 | 85.0 | 65.9 | 58.2 |
| | ✓ | **71.9** | **85.8** | **74.0** | **68.7** |
| VLsI-7B | ✗ | 74.7 | 85.8 | 75.2 | 69.3 |
| | ✓ | **76.8** | **86.2** | **81.2** | **73.4** |

As shown in Table 5(a), a single update iteration ($\mu = 1$) for both is sufficient for strong performance; more iterations, especially for student VLM updates, can risk overfitting. Subsequently, we explore the impact of the KL-divergence penalty coefficient $\beta$ (Eq. 2) in Table 5(b). These results confirm that while constraining policy updates is crucial for stability, an overly restrictive penalty (large $\beta$) can hinder performance.

Next, we assess decisions related to model component updates. Table 5(c) details the importance of updating different parameter groups within the student VLMs, revealing that training the self-attention layers, word embeddings, and the language head yields more significant gains than updating feed-forward networks (FFN) or layer normalization parameters. We also investigate the necessity of continuously updating the discriminator during RIL training (Table 5(d)). Interestingly, Table 5(d) shows that using a fixed (not updated: △), pre-trained discriminator can outperform RL-only baselines (denoted by ✗), but overall RIL efficacy still critically depends on leveraging a continuously trained discriminator (denoted ✓) during training RIL.

In addition, we evaluate RIL's interaction with evaluation methodologies of the predicted answer and its synergy with other training paradigms. Table 5(e) underscores the importance of LLM-as-a-Judge [9] for answer rewards; replacing it with conventional answer parsing (denoted △) significantly degrades performance, particularly on general visual questions (e.g., "How to cook this dish?" from an image) where fixed parsing rules are inadequate. This highlights the need for flexible, LLM-based evaluation. The same table also affirms that while initial SFT is beneficial for capturing the training data distribution, RIL subsequently delivers substantially larger improvements.

On Table 5(f), we examine RIL's effectiveness on student VLMs that are already trained on distillation methods like MiniLLM [75], DistilLLM [76], LLaVA-KD [77], and VLsI [45] (on same backbones like Qwen2-VL [16] and training datasets C). As shown in Table 5(f), RIL proves particularly potent for these models compared to non-distillation counterparts, likely because prior feature alignment through distillation primes them for RIL's alignment mechanisms.

Lastly, we demonstrate that the binary similarity reward introduced in Sec. 3.3 enables more efficient and stable training of RIL. As shown in Fig. 6, the binary reward consistently outperforms continuous and multi-level discretizations, suggesting that binary feedback provides clearer and more reliable learning signals. This improvement arises because models often struggle to interpret subtle differences in continuous rewards (e.g., why a score of 0.21 should be preferred over 0.20).

## 4.3 Discussion and Limitation

RIL leverages the diversity of responses generated by large VLMs, which tend to produce more varied and higher-quality answers beyond the ground truth answers used in SFT training datasets. As illustrated in Fig. 7, increasing the number of teacher-generated responses consistently improves the student model's performance, demonstrating that richer supervision from diverse teacher outputs enhances the student VLM's overall generalization ability. In summary, RIL exploits these diverse, high-quality teacher responses to guide smaller VLMs toward learning more flexible responses beyond fixed ground truth answers.

Furthermore, RIL offers distinct advantages over traditional knowledge distillation techniques [78]. Unlike conventional distillation, which typically requires student VLMs to replicate the high-

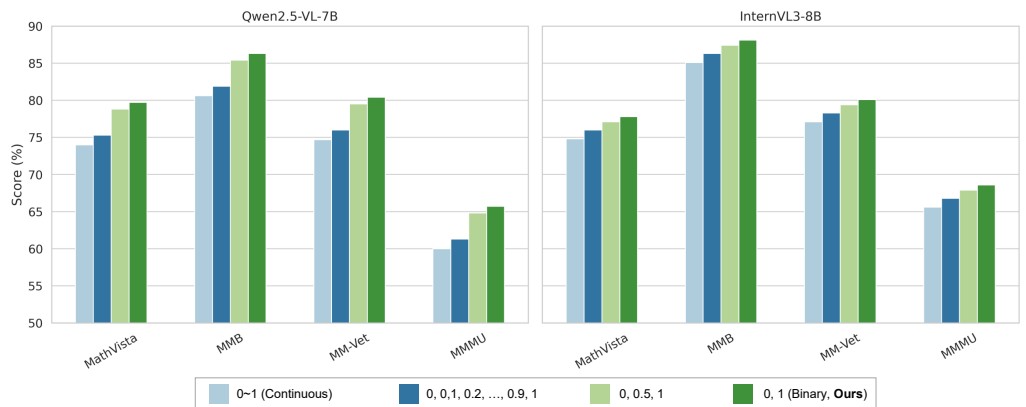

Figure 6: Comparison of RIL performance using different discretization levels of the similarity reward on four benchmarks: MathVista [3], MMB [20], MM-Vet [4], and MMMU [5]) with Qwen2.5-VL-7B [1] and InternVL3-8B [18].

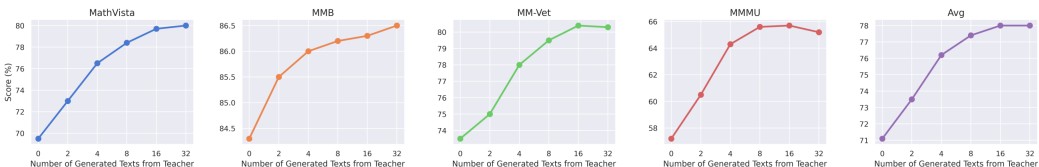

Figure 7: Performance of RIL with varying numbers of generated responses from a teacher VLM across four benchmarks and their average. They are evaluated on Qwen2.5-VL-7B [1].

dimensional logit features of teacher models, RIL operates exclusively on generated text responses in natural language. This fundamental difference liberates RIL from restrictive feature-level constraints, such as the need for identical image embedding strategies or perfectly aligned language tokenizers (including vocabulary, index order, and sequence length) between student and teacher VLMs. Common distillation methods [75–77, 45] often rely on KL divergence—a static, non-trainable metric devoid of contextual understanding—to measure feature similarity, but RIL employs a trainable discriminator which assesses natural language responses directly, learning to discern nuanced similarities and stylistic differences between student and teacher outputs. The adaptability of the trainable discriminator in capturing contextual relationships makes RIL a particularly flexible and powerful approach for lightweight VLM and enhancement.

While RIL demonstrates these significant strengths, its current implementation has been focused primarily on the post-instruction tuning alignment phase. A promising avenue for future research, therefore, involves extending the utility of our discriminator to the initial visual instruction tuning stage itself. We hypothesize that the discriminator's ability to learn from natural language signals and overcome conventional architectural constraints could also offer significant benefits in this earlier phase of VLM training, potentially leading to more efficient instruction following from the outset.

## 5 Conclusion

In this work, we have introduced Unified Reinforcement and Imitation Learning (RIL), a novel framework enabling smaller VLMs to emulate the sophisticated text-generation capabilities of substantially larger counterparts, often matching or even exceeding their performance. RIL directly addresses the pressing need for high-performing, lightweight VLMs deployable in resource-constrained settings. By leveraging a dual reward system—combining similarity scores from a discriminator with answer accuracy assessments from an LLM-as-a-Judge—RIL effectively guides student VLMs towards superior alignment and overall efficacy. Key findings from our research include the significant performance boost achieved by using multiple large teacher VLMs, which provides a richer diversity of training signals, and the notable success of RIL in enhancing distillation-based VLMs. Comprehensive experiments across a wide array of vision-language benchmarks confirm that RIL substantially narrows, and in several cases surpasses, the performance gap to state-of-the-art open- and closed-source VLMs.

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

# A  Algorithm of RL only with Answer Rewards from LLM-as-a-Judge

---

**Algorithm 1** RL purely with GRPO or Dr.GRPO based on accuracy rewards from LLM-as-a-Judge

---

**Require:** Pre-trained VLMs $\pi_{\theta_{\text{init}}}$
 1: Set reference model $\pi_{\text{ref}} \leftarrow \pi_{\theta_{\text{init}}}$
 2: Set the training model $\pi_\theta \leftarrow \pi_{\theta_{\text{init}}}$
 3: **for** sample a batch $\mathcal{B}$ in Dataset **do**
 4:    Copy and freeze model $\pi_{\theta_{\text{old}}} \leftarrow \pi_\theta$
 5:    Sample $G$ outputs $\{o_i\}_{i=1}^G \sim \pi_{\theta_{\text{old}}}(\cdot|q)$ for each question $q \in \mathcal{B}$
 6:    Compute Rewards and Advantages for all $G$ outputs by LLM-as-a-Judge
 7:    **for** VLMs Updating iteration $= 1, 2, \cdots, \mu$ **do**
 8:       Update $\pi_\theta$ by using the objective of GRPO [43] or Dr.GRPO [8]
 9:    **end for**
10: **end for**

---

# B  Algorithm of RIL for VLMs

---

**Algorithm 2** RIL for VLMs

---

**Require:** Pre-trained discriminator $D_\phi$ and Pre-trained student VLMs $\pi_{\theta_{\text{init}}}$
**Require:** Saved collection for the generated text responses $\{o^{(l)}\}$ from teacher VLMs
 1: Set reference model $\pi_{\text{ref}} \leftarrow \pi_{\theta_{\text{init}}}$
 2: Set the training model $\pi_\theta \leftarrow \pi_{\theta_{\text{init}}}$
 3: **for** sample a batch $\mathcal{B}$ in Dataset **do**
 4:    Copy and freeze model $\pi_{\theta_{\text{old}}} \leftarrow \pi_\theta$
 5:    Sample $G$ outputs $\{o_i^{(s)}\}_{i=1}^G \sim \pi_{\theta_{\text{old}}}(\cdot|q)$ for each question $q \in \mathcal{B}$
 6:    Extract $G$ outputs $\{o_i^{(t)}\}_{i=1}^G$ in the saved collection for each question $q \in \mathcal{B}$
 7:    **for** Discriminator Updating iteration $= 1, 2, \cdots, \mu$ **do**
 8:       Update $D_\phi$ by using Equation 1
 9:    **end for**
10:    Compute Rewards and Advantages for all $2G$ outputs by Discriminator and LLM-as-a-Judge
11:    **for** Student VLMs Updating iteration $= 1, 2, \cdots, \mu$ **do**
12:       Update $\pi_\theta$ by using Equation 2
13:    **end for**
14: **end for**

---

# C  Implementation Detail

**Details of Training and Evaluation.**   We train and evaluate RIL, mainly on NVIDIA A100 80GB GPUs. Because it is practically critical point to get fast text generation during training, we utilize vLLM [79] built on PagedAttention. For the pre-training step of discriminator, we assign vLLM [79] to 8 GPUs for fast text generation, and we generate $N$=16 text responses for each question at student and teacher VLMs, respectively. Once it is finished, we use DeepSpeed engine with ZeRO-3 [80] for 8 GPUs, and we use AdamW optimizer [81] and apply a linearly decayed learning rate from 1e-5 to 1e-6 to pre-training discriminator and SFT of student VLMs. In subsequent step, when training both discriminator and student VLMs, one of 8 GPUs is assigned to conduct online generation of student VLMs by vLLM [79]. In addition, another GPU is assigned to conduct LLM-as-a-Judge [9] by vLLM [79]. The other 6 GPUs are assigned to use DeepSpeed engine with ZeRO-3 [80]. Mimicking step requires static learning rate 1e-6 and $\mu$=1 iteration to train both student VLMs and the discriminator. Note that, when we generate text responses, we generate $G$=4 responses for each question, by setting temperature to 1.0, top-p to 0.95, top-k to 50, and repetition penalty to 1.05, in order to get diverse text responses. For stable training, we handle large batch sizes by using gradient accumulation with 6 steps. At every step, we use 4 batches per one GPU, leading to total 144 batches. For evaluation, we use distributed data parallel to load student VLMs for all 8 GPUs and

score the correctness or generation quality of their text responses by using their default generation hyperparameter.

**Computational Cost.**  Compared to GRPO [43], RIL requires increased computational costs during training due to teacher, student, discriminator, and LLM-as-a-Judge [9]. However, we would like to emphasize that discriminator and student costs can be at least mitigated. While pretraining the discriminator is a necessary step and cannot be avoided from a computational standpoint, we can reduce training costs of discriminator and student during the RIL loop (Algorithm 2) by managing model weights efficiently between CPU and GPU. Specifically, we offload the weights of the discriminator and student to the CPU when they are not in use. When needed, we load the appropriate weights from the CPU to the GPU using DeepSpeed API [80]. This allows us to load or unload model weights without uploading both architecture to GPU, thereby minimizing computational overhead. This approach is feasible because we use the same model architecture for both the discriminator and student, enabling seamless weight replacement and resource-efficient training.

**Visual Instruction Tuning Dataset.**  We assemble a 4M-sample visual instruction tuning dataset that encompasses a diverse array of vision-language tasks, including general visual question answering, dense image captioning, chart, diagram, and document understanding, common-sense knowledge, scientific and mathematical problem-solving, and multidimensional reasoning. For SFT, we utilize the entire 4M-sample dataset, and then we curate a 40K-sample dataset for RIL of VLMs based on log-probability sampling [82] and overlong filtering [36]. Our dataset integrates both real-world and synthetic sources: COCO-ReM [83], iNaturalist2018 [84], VQA-v2 [85], Super-CLEVR [86], MAVIS [87], Geometry3K [88], SQA [89], AI2D [2], SA-1B [60], LLaVAR [90], VSR [91], TallyQA [92], TabMWP [93], KonIQ [94], InternVL [95]-filtered synthetic knowledge dataset covering politics, math, physics, chemistry, RLAI-F [96], CLEVR-Math [97], SROIE [98], ChartQA [19], DocVQA [99], FigureQA [100], GQA [101], InfoVQA [102], M3CoT [103], MapQA [104], OK-VQA [105], TextVQA [106], WildVision [107], DVQA [108], GeoQA+ [109], GeoS [110], IconQA [111], UniGEO [112], GeomVerse [113], Geo170K [114], MathV360K [115], multimodal wikipedia knowledge [116], InfoSeek [117], and RAM++ [118]-filtered synthetic data of Infinity-MM [30] covering coarse and fine-grained perception, relation, attribute, and logic reasoning.

# D   Broader Impacts

Our reinforcement and imitation learning (RIL) approach democratises access to advanced vision–language capabilities by enabling lightweight, efficient models to run on edge devices and in low-resource settings—thereby lowering both deployment costs and environmental impact. RIL fosters seamless interoperability across diverse model architectures and tokenization schemes. The use of multiple large teacher VLMs to generate varied training samples further accelerates performance gains, underscoring the value of collaborative model development. We believe RIL will inspire future research into efficient multimodal learning techniques and support the widespread, sustainable adoption of lightweight [119] and robust [120–122] AI solutions across a broad range of applications even with different tokenizers [123]. We hope that by bridging efficiency and performance, RIL become a landscape of building sophisticated multimodal AI technologies and accelerating innovation across industries and academia.

