# OpenReview forum: "Unified Reinforcement and Imitation Learning for Vision-Language Models"
_NeurIPS.cc/2025/Conference — NeurIPS 2025 poster_

### Official Review · Reviewer_2zPb · 2025-06-28

**Clarity:** 3
**Significance:** 3
**Originality:** 2
**Rating:** 4
**Confidence:** 4

**Summary:**

This paper presents a novel training framework, Unified Reinforcement and Imitation Learning (RIL), designed for efficient and lightweight Vision-Language Models (VLMs). By integrating reinforcement learning (RL) and adversarial imitation learning (GAIL), RIL enables smaller "student" VLMs to mimic sophisticated text-generation patterns of larger "teacher" models and simultaneously improve through reinforcement signals. A key innovation is the use of a large language model (LLM)-based discriminator to distinguish between teacher and student outputs, combined with an LLM-as-a-Judge system to verify factual correctness, ensuring both stylistic and factual alignment. Experiments demonstrate that RIL significantly improves student model performance, achieving results competitive with or superior to leading open- and closed-source models.

**Questions:**

1. What mechanisms or hyperparameter tuning strategies are critical to maintaining training stability, particularly in adversarial imitation settings?
2. Could RIL be effectively combined with other recent model alignment methods (e.g., instruction tuning or retrieval augmentation)?
3. How sensitive is the approach to the selection and quality of large teacher VLMs? Would performance degrade substantially if lower-quality teachers were used?
4. In the manuscript, "Eq. (x)" and "Eq. x" are used interchangeably. Please standardize their usage throughout the manuscript.
5. If my concerns are addressed, I will increase my score after the rebuttal.

**Ethical Concerns:**

["NO or VERY MINOR ethics concerns only"]

**Limitations:**

See Questions.

**Paper Formatting Concerns:**

None.

**Quality:**

3

**Strengths And Weaknesses:**

Strengths:
1. The unified use of reinforcement learning and adversarial imitation learning is novel and effective.
2. The incorporation of an LLM-based discriminator and a judge-model (LLM-as-a-Judge) enhances training robustness.
3. Extensive evaluation across diverse vision-language benchmarks (AI2D, MathVista, MM-Vet, MMMU, BLINK, etc.) confirms the superiority of RIL-trained models compared to state-of-the-art baselines.
4. Well-written with clear explanations of methodology, extensive experiments, ablation studies, and detailed training procedures.

Weaknesses:
1. The requirement of multiple large VLMs and LLMs for both discriminator and judge-model training might limit accessibility due to significant computational resource demands.
2. This manuscript is well-written and presents a strong empirical contribution to efficient vision-language modeling. However, the degree of innovation might appear incremental given the existing frameworks. Please further emphasize the innovation and the problems addressed by this paper.

---

> ### Author Rebuttal · Authors · 2025-07-26
>
> **Q1. The requirement of multiple large VLMs and LLMs for both discriminator and judge-model training might limit accessibility due to significant computational resource demands.**
>
> Compared to the original rule-based GRPO, we agree that RIL introduces increased computational costs during training. However, we would like to emphasize that these costs can be mitigated through efficient technical implementation strategies. While pretraining the discriminator is a necessary step and cannot be avoided from a computational standpoint, we can reduce training costs during the RIL loop by managing model weights efficiently. Specifically, we offload the weights of the small VLMs and the discriminator to the CPU when they are not in use. When needed, we load the appropriate weights from the CPU to the GPU using DeepSpeed API. This allows us to dynamically swap model weights without uploading both architecture to GPU, thereby minimizing computational overhead. This approach is feasible because we use the same model architecture for both the small VLMs and the discriminator, enabling seamless weight replacement and resource-efficient training.
>
> ---
>
> **Q2. This manuscript is well-written and presents a strong empirical contribution to efficient vision-language modeling. However, the degree of innovation might appear incremental given the existing frameworks. Please further emphasize the innovation and the problems addressed by this paper.**
>
> We would like to emphasize two key points regarding the innovation of our work:
>
> - While individual components such as Dr.GRPO and LLM-as-a-Judge have been previously explored, the overall framework we propose is novel in that it enables the distillation of knowledge from multiple large VLMs into a single, smaller VLM. This unified design allows us to leverage the strengths of various large models in a systematic and scalable manner.
>
> - Our approach is orthogonal to advancements in VLM architectures and agnostic to specific VLM series, thanks to the generalized nature of our framework. This makes it broadly applicable and future-proof as new VLMs emerge.
>
> Specifically, RIL provides a generalization effect similar to RL and mimics the benefits of high-quality SFT, guiding smaller VLMs to learn more flexible responses beyond fixed ground-truth answers. This is because RIL basically leverages the diversity of responses generated by large VLMs, which are more likely to produce high-quality answers compared to smaller VLMs, and these generated responses are directly used to simultaneously train both the discriminator and the smaller VLMs, as described in lines 6–13 of Algorithm 2.
>
> Thanks to such sophisticated framework design, the experimental results showed the significant performance improvement on multiple benchmarks with multiple VLM families. Therefore, once the new VLM is developed in the community, this approach can exploit such model to develop lighter and more accurate VLMs.
>
> We will add this explanation to the manuscript to better clarify this important point.
>
> ---
>
> **Q3. What mechanisms or hyperparameter tuning strategies are critical to maintaining training stability, particularly in adversarial imitation settings?**
>
> We have addressed the training stability from multiple perspectives:
>
> - Structured Prompt for LLM-as-a-Judge: Please refer to our response of Reviewer SQq6's Q3.
> - Binary Answer Reward: Please refer to our response of Reviewer SQq6's Q3.
> - Binary Similarity Reward: Please refer to our response of Reviewer Hg1D’s Q4.
> - Quality and Diversity of Generated Responses from Teacher Models: Please refer to our response of Reviewer SQq6’s Q4.
> - Hyperparameter: To prevent unstable training, we conducted several ablation studies on the number of discriminator and small VLM training iterations, as well as on the KL divergence hyperparameter, as shown in Table 5(a) and (b).
>
> ---
>
> **Q4. Could RIL be effectively combined with other recent model alignment methods (e.g., instruction tuning or retrieval augmentation)?**
>
> We have already discussed RIL's limitations in the discussion section (lines 285–291) on page 9, noting that the current implementation of RIL primarily focuses on the post-instruction tuning alignment phase. Extending RIL to earlier stages, such as the visual instruction tuning phase, would be a promising direction for future research.
>
> ---
>
> **Q5. How sensitive is the approach to the selection and quality of large teacher VLMs? Would performance degrade substantially if lower-quality teachers were used?**
>
> Building on our answer to Reviewer SQq6’s Q4, we have conducted additional experiments based on your comment regarding the use of lower-quality teachers.
>
> Both the diversity and the quality of teacher outputs are critical to RIL’s success. As the ablation study below shows, RIL’s performance degrades when teacher responses lack diversity or accuracy. (We assume diversity is satisfied by generating a larger number of responses, since we set the inference hyperparameters—temperature, top‑p, and top‑k.)
>
> The first table reports performance as we vary the number of pre‑generated teacher responses used to train the discriminator and drive the RIL loop:
>
> | Number of Generated Responses | MathVista | MMB  | MM-Vet | MMMU | Avg  |
> |-------------------------------|:-----------:|:------:|:--------:|:------:|:------:|
> | 0 (Dr.GRPO)                   | 69.5      | 84.3 | 73.5   | 57.2 | 71.1 |
> | 2                             | 73.0      | 85.5 | 75.0   | 60.5 | 73.5 |
> | 4                             | 76.5      | 86.0 | 78.0   | 64.3 | 76.2 |
> | 8                             | 77.0      | 86.2 | 79.0   | 68.6 | 77.7 |
> | 16                            | 79.7      | 86.3 | 80.4   | 65.7 | 78.0 |
> | 32                            | 80.0      | 86.5 | 80.3   | 65.2 | 78.0 |
>
> Note that, this result is based on Qwen2.5-VL-RIL-7B (Both).
>
> We also evaluated the effect of introducing incorrect answers by randomly shuffling a percentage of generated responses relative to their questions (a dash “–” indicates mode collapse):
>
> | Shuffle Percent  | MathVista | MMB  | MM-Vet | MMMU | Avg  |
> |------------------|:-----------:|:------:|:--------:|:------:|:------:|
> | 0\%              | 79.7      | 86.3 | 80.4   | 65.7 | 78.0 |
> | 10\%             | 75.0      | 82.0 | 77.0   | 59.6 | 73.4 |
> | 30\%             | 62.0      | 68.0 | 60.0   | 30.0 | 55.0 |
> | 50\%             | -         | -    | -      | -    | -    |
> | 100\%            | -         | -    | -      | -    | -    |
>
> Note that, this result is based on Qwen2.5-VL-RIL-7B (Both).
>
> Additionally, we evaluated the impact of using downgraded versions of "Qwen2-VL-72B" and "InternVL2.5-78B" as lower-quality teachers on RIL’s performance:
>
> | Models                                               | MathVista | MMB  | MM-Vet | MMMU | Avg  |
> |------------------------------------------------------|:-----------:|:------:|:--------:|:------:|:------:|
> | Qwen2.5-VL-RIL-7B (Qwen2-VL-72B and InternVL2.5-78B) | 70.5      | 84.2 | 76.1   | 62.6 | 73.3 |
> | Qwen2.5-VL-RIL-7B (Qwen2.5-VL-72B and InternVL3-78B) | 79.7      | 86.3 | 80.4   | 65.7 | 78.0 |
> | InternVL3-RIL-8B (Qwen2-VL-72B and InternVL2.5-78B)  | 72.0      | 86.0 | 77.4   | 67.2 | 75.7 |
> | InternVL3-RIL-8B (Qwen2.5-VL-72B and InternVL3-78B)  | 77.8      | 88.1 | 80.1   | 68.6 | 78.7 |
>
>
> **Recap**: Based on these results and analyses, we argue that our approach is orthogonal to advancements in VLM families and agnostic to VLM series, due to the generalized nature of our framework. Therefore, as new VLMs are developed by the community, our approach can seamlessly incorporate them to train smaller and more accurate VLMs.
>
> In summary,
>
> - RIL benefits substantially from both diverse and accurate teacher outputs.
> - Performance improves steadily as the number of high-quality, varied responses increases.
> - Conversely, introducing inconsistent or conflicting answers causes significant performance degradation.
> - The use of lower-quality teachers leads to noticeably reduced performance.
>
> ---
>
> **Q6. In the manuscript, "Eq. (x)" and "Eq. x" are used interchangeably. Please standardize their usage throughout the manuscript.**
>
> We have identified that "Eq. x" appears on lines 205 and 250. We will revise these instances to "Eq. (x)" to ensure consistency throughout the manuscript.
>
> ---
>
> Thank you for your encouraging review. We hope our responses have further clarified the points you raised.

---

### Official Review · Reviewer_oXVe · 2025-06-29

**Clarity:** 4
**Significance:** 3
**Originality:** 4
**Rating:** 5
**Confidence:** 5

**Summary:**

This paper introduces a new approach for combining reinforcement learning and imitation learning to improve the reasoning capabilities of Vision-Language Models (VLMs). The core idea is to use a larger teacher VLM whose responses guide the post-training of a smaller VLM through imitation. This motivation is supported by improved performance observed under this training paradigm. The authors present their findings with clear experiments and well-designed ablations that effectively support the proposed method.

**Questions:**

* Based on the experiments, it appears that applying SFT before RIL improves post-training outcomes. In Table 1, when comparing against the GRPO baseline, could the authors clarify whether the GRPO baseline also included SFT as a prior step?
* I understand the single-teacher LLM setup, where the discriminator model provides a binary signal indicating whether a response comes from the teacher. Could the authors clarify how this approach extends to the multiple-teacher scenario?
* Finally, could the authors provide a comparison of the speed to achieve good performance versus vanilla GRPO? It would be interesting to see whether the proposed RIL approach leads to faster convergence.

**Ethical Concerns:**

["NO or VERY MINOR ethics concerns only"]

**Final Justification:**

The paper is technically solid and proposes a novel way to combine imitation learning (mimicing a teacher) and RL to improve VLMs. Given the experiments and the analyses performed I would stick with my original rating of 'Accept'

**Limitations:**

Yes

**Quality:**

4

**Strengths And Weaknesses:**

**Strengths**

* The paper is clearly written and easy to follow, with strong experimental support and well-designed ablations for the proposed method.
* The idea of combining reinforcement learning and imitation learning is well established in traditional RL literature, and adapting this approach to VLMs is both novel and interesting.

**Weaknesses**

* One limitation is that the paper focuses solely on VLMs; it would have been valuable to see the approach extended to LLMs as well.
* Given the large number of tasks in Table 1, it would be helpful if the authors included an average metric to make it easier to assess the overall improvement at a glance.

---

> ### Author Rebuttal · Authors · 2025-07-26
>
> **Q1. One limitation is that the paper focuses solely on VLMs; it would have been valuable to see the approach extended to LLMs as well.**
>
> We have also applied RIL to LLMs to evaluate its generalization capabilities. Based on these results, we confirm that RIL is effective for LLMs as well.
>
> | Student LLMs (Teacher LLMs)                        | MMLU | GPQA | MATH |
> |------------------------------|:------:|:------:|:------:|
> | Qwen2.5-7B                   | 74.2 | 36.4 | 49.8 |
> | Qwen2.5-7B-RIL (Qwen2-72B)   | 79.6 | 37.0 | 50.4 |
> | Qwen2.5-7B-RIL (Qwen2.5-72B) | 81.0 | 41.2 | 56.7 |
> | Qwen2-72B                    | 84.2 | 37.4 | 50.9 |
> | Qwen2.5-72B                  | 86.1 | 45.9 | 62.1 |
>
> Note that, rows without parentheses indicate baseline models without any distillation.
>
> ---
>
> **Q2. Given the large number of tasks in Table 1, it would be helpful if the authors included an average metric to make it easier to assess the overall improvement at a glance.**
>
> We will definitely include an average metric to facilitate easier assessment of overall improvements at a glance. As shown in the table below, it is clear that the proposed approach, "w. RIL (Dr.GRPO + GAIL)", achieves the best performance across all VLM families—even when using only a single teacher VLM.
>
> | VLMs                    | Avg   |
> |-------------------------|:-------:|
> | Qwen2.5-VL-7B           | 69.4  |
> | w. RL (GRPO)            | 70.3  |
> | w. RL (Dr.GRPO)         | 71.1  |
> | w. RIL (Dr.GRPO + GAIL) | 75.1  |
> | Qwen2.5-VL-3B           | 64.1  |
> | w. RL (GRPO)            | 65.5  |
> | w. RL (Dr.GRPO)         | 66.4  |
> | w. RIL (Dr.GRPO + GAIL) | 68.7  |
> | InternVL3-8B            | 71.4  |
> | w. RL (GRPO)            | 72.5  |
> | w. RL (Dr.GRPO)         | 73.1  |
> | w. RIL (Dr.GRPO + GAIL) | 75.3  |
> | InternVL3-2B            | 62.6  |
> | w. RL (GRPO)            | 63.6  |
> | w. RL (Dr.GRPO)         | 64.5  |
> | w. RIL (Dr.GRPO + GAIL) | 66.1  |
> | InternVL3-1B            | 55.7  |
> | w. RL (GRPO)            | 56.5  |
> | w. RL (Dr.GRPO)         | 57.2  |
> | w. RIL (Dr.GRPO + GAIL) | 59.8  |
>
> ---
>
> **Q3. Based on the experiments, it appears that applying SFT before RIL improves post-training outcomes. In Table 1, when comparing against the GRPO baseline, could the authors clarify whether the GRPO baseline also included SFT as a prior step?**
>
> Yes. All experiments reported in the manuscript, including the GRPO baseline, were conducted after completing the SFT training phase.
>
> ---
>
> **Q4. I understand the single-teacher LLM setup, where the discriminator model provides a binary signal indicating whether a response comes from the teacher. Could the authors clarify how this approach extends to the multiple-teacher scenario?**
>
> Given a response T1 from one teacher VLM and T2 from another teacher VLM, and a response S from the student VLM, then the discriminator is trained to output zero when S is provided and one when either T1 or T2 is provided. Additionally, the rewards and their advantages for S, T1, and T2 are computed based on similarity and answer quality. The student VLM is updated using RIL’s objective loss. We will clarify the multi-teacher training setup in the revised manuscript.
>
> ---
>
> **Q5. Finally, could the authors provide a comparison of the speed to achieve good performance versus vanilla GRPO? It would be interesting to see whether the proposed RIL approach leads to faster convergence.**
>
> In terms of convergence speed, GRPO actually converges faster than the RIL framework. This is because GRPO does not rely on any external models, such as a discriminator or a teacher model, but instead improves upon itself. This self-reliance imposes a natural limitation on performance improvement, making the convergence appear relatively quick. In contrast, RIL leverages external models to guide the learning process, allowing it to gradually improve and ultimately surpass the performance of GRPO. While this results in slower convergence compared to GRPO, the overall performance gain is significantly higher, as demonstrated across multiple experiments. (\% denotes training percentage, and the results describe average metrics for 14 evaluation benchmarks based on Table 1 in the manuscript.)
>
> | Methods | 0\%  | 10\%| 20\%| 30\%| 40\%| 50\%| 60\%| 70\%| 80\%| 90\%| 100\%|
> |---------|:------:|:-----:|:-----:|:-----:|:-----:|:-----:|:-----:|:-----:|:-----:|:-----:|:------:|
> | GRPO    | 69.7 | 69.9| 70.0| 70.1| 70.2| 70.3| 70.3| 70.3| 70.3| 70.3| 70.3 |
> | Dr.GRPO | 69.7 | 70.0| 70.3| 70.7| 70.8| 71.0| 71.1| 71.1| 71.1| 71.1| 71.1 |
> | RIL (Single Teacher)     | 69.7 | 69.9| 70.3| 71.2| 71.5| 72.6| 74.0| 74.7| 75.0| 75.1| 75.1 |
>
> Note that, this result is based on Qwen2.5-VL-7B.
>
> ---
>
> Thank you for your encouraging review. We hope our responses have further clarified the points you raised.

---

> > ### Comment · Reviewer_oXVe · 2025-08-04
> > **Response to Authors**
> >
> > Thanks for clarifying my additional questions. I would also recommend to add the new experiments and analyses discussed here as it offers additional insight about the paper, especially the experiments on LLMs and convergence analysis.
> >
> > Since I already gave a good rating, I will keep that post rebuttal
> >
> > Thank you.

---

> > > ### Author Response · Authors · 2025-08-04
> > >
> > > Thank you for your kind and constructive feedback. We're glad the additional clarifications were helpful, and we appreciate your positive rating and thoughtful suggestions regarding the new experiments.

---

### Official Review · Reviewer_SQq6 · 2025-07-01

**Clarity:** 3
**Significance:** 3
**Originality:** 2
**Rating:** 4
**Confidence:** 4

**Summary:**

This paper presents Unified Reinforcement and Imitation Learning, a novel training framework designed to enhance small VLMs by enabling them to imitate the performance and generation style of significantly larger models. RIL integrates reinforcement learning with adversarial imitation learning, allowing student models to optimize for both stylistic similarity (to large teacher models) and factual correctness (via LLM-as-a-Judge evaluations). Extensive experiments on a variety of benchmarks demonstrate that student VLMs trained with RIL can not only match but sometimes outperform state-of-the-art open- and closed-source models, despite their relatively smaller sizes.

**Questions:**

1. How sensitive is RIL to the quality or diversity of teacher outputs? Would performance degrade if teacher models produced inconsistent or conflicting answers?

2. Are there any observed failure modes (e.g., mode collapse, overfitting to teacher phrasing) during RIL training? If so, how are they mitigated?

3. How is the answer parsed from the responses generated by RIL-finetuned models, given that no specific response format is enforced? For instance, ChartQA requires short phrase answers, while several other benchmarks demand a discrete choice selection.

4. What prompt format is used for the teacher models? For questions with deterministic answers, the similarity and answer rewards may become highly correlated, potentially limiting the extent of knowledge transfer from teacher to student.

**Ethical Concerns:**

["NO or VERY MINOR ethics concerns only"]

**Final Justification:**

Most of my issues have been addressed after the authors included additional sensitive experiments. As a result, I raise my score to borderline accept. That said, the paper still requires major revisions, particularly in the areas of computational demands and case study. I recommend that the authors revise these aspects if the paper is accepted. However, rejecting the paper and reevaluating it through a new submission would also be acceptable.

**Limitations:**

yes

**Quality:**

3

**Strengths And Weaknesses:**

**Strengths**

1. Innovative Integration: The paper successfully combines reinforcement and imitation learning in a unified framework, addressing limitations in prior work that rely solely on one learning paradigm.

2. Strong Empirical Results: The method demonstrates state-of-the-art or competitive performance across numerous benchmarks and significantly narrows the gap between small and large VLMs.

3. Robustness Across Models: RIL is shown to work with different student and teacher architectures and is particularly effective when multiple teachers are used.

**Weaknesses**

1. Limited Discriminator Design: As shown in Figure 3, the discriminator does not receive any information about the question. Despite fine-tuning, it relies solely on limited cues from the response, and no examples of its decision outputs are provided.

2. Increased Complexity and Cost: Unlike the original GRPO method that employs rule-based rewards, the proposed approach relies on multiple models to compute rewards. Some components, such as the discriminator, require pretraining or continual fine-tuning, increasing both complexity and computational cost.

3. Insufficient Discussion of LLM Bias and Reward Hacking: The core idea of using multiple LLMs as evaluators introduces risks of bias, (e.g., https://arxiv.org/html/2410.21819v1 ). Moreover, compared to rule-based rewards, LLM-based evaluation is more susceptible to reward hacking, a concern not adequately addressed in the paper.

---

> ### Author Rebuttal · Authors · 2025-07-26
>
> **Q1. The discriminator relies solely on limited cues from the response, and no examples of its decision outputs are provided.**
>
> While the discriminator only receives the response as input, this design is intentional for the sake of training efficiency. We have observed that incorporating additional input cues (such as the question including image) does not consistently improve performance and instead introduces more randomness. This suggests that the key contribution lies not in carefully tuning the input format, but in leveraging the discriminator itself as a training signal. Therefore, to reduce computational cost, we deliberately shorten the discriminator's input. For greater clarity, we will definitely include qualitative examples (image, question, predicted response, decision) of the discriminator's outputs in the revised version.
>
> | Models        | Discriminator Input                  | MathVista | MMB  | MM-Vet | MMMU | Avg  |
> |---------------|--------------------------------------|:-----------:|:------:|:--------:|:------:|:------:|
> | Qwen2.5-VL-7B | Question + Response                  | 79.0      | 85.5 | 80.2   | 66.5 | 77.8 |
> | Qwen2.5-VL-7B | Response                             | 79.7      | 86.3 | 80.4   | 65.7 | 78.0 |
> | InternVL3-8B  | Question + Response                  | 78.5      | 88.0 | 81.0   | 67.7 | 78.8 |
> | InternVL3-8B  | Response                             | 77.8      | 88.1 | 80.1   | 68.6 | 78.7 |
>
> ---
>
> **Q2. Increased Complexity and Cost**
>
> Compared to the original rule-based GRPO, we agree that RIL introduces increased computational costs during training. However, we would like to emphasize that these costs can be mitigated through efficient technical implementation strategies. While pretraining the discriminator is a necessary step and cannot be avoided from a computational standpoint, we can reduce training costs during the RIL loop by managing model weights efficiently. Specifically, we offload the weights of the small VLMs and the discriminator to the CPU when they are not in use. When needed, we load the appropriate weights from the CPU to the GPU using DeepSpeed API. This allows us to dynamically swap model weights without uploading both architecture to GPU, thereby minimizing computational overhead. This approach is feasible because we use the same model architecture for both the small VLMs and the discriminator, enabling seamless weight replacement and resource-efficient training.
>
> ---
>
> **Q3. Insufficient Discussion of LLM Bias and Reward Hacking**
>
> To mitigate reward hacking and bias when using LLM-as-a-Judge, we employ a structured, template‑based format: we wrap the user query, the ground truth, and the model’s output in question, ground truth, and generated text tags, respectively. It is then prompted to explain its reasoning and generate its judgment within answer tags. This structured approach has been widely adopted in prior work [1–5]. Additionally, we cut off the answer reward signal of LLM-as-a-Judge to avoid overestimation or reward hacking, which leads to more stable training. The table below shows how performance varies with different granularities of the answer reward, where a dash '–' denotes mode collapse—training failed because small VLMs could not reliably distinguish, for instance, an answer reward of '0.2' from '0.21'.
>
>
> | Models        | Reward Step (Answer Reward)                       | MathVista | MMB  | MM-Vet | MMMU | Avg  |
> |---------------|---------------------------------------------------|:-----------:|:------:|:--------:|:------:|:------:|
> | Qwen2.5-VL-7B | 0~1 (Continuous)                                  | -         | -    | -      | -    | -    |
> | Qwen2.5-VL-7B | 0, 0.1, 0.2, 0.3, 0.4, 0.5, 0.6, 0.7, 0.8, 0.9, 1 | 60.2      | 78.2 | 64.2   | 50.9 | 63.4 |
> | Qwen2.5-VL-7B | 0, 0.5, 1                                         | 74.9      | 84.3 | 76.4   | 62.0 | 74.4 |
> | Qwen2.5-VL-7B | 0,1 (Binary)                                      | 79.7      | 86.3 | 80.4   | 65.7 | 78.0 |
> | InternVL3-8B  | 0~1 (Continuous)                                  | -         | -    | -      | -    | -    |
> | InternVL3-8B  | 0, 0.1, 0.2, 0.3, 0.4, 0.5, 0.6, 0.7, 0.8, 0.9, 1 | 64.8      | 76.7 | 71.4   | 55.9 | 67.2 |
> | InternVL3-8B  | 0, 0.5, 1                                         | 73.8      | 84.6 | 77.4   | 64.7 | 75.1 |
> | InternVL3-8B  | 0,1 (Binary)                                      | 77.8      | 88.1 | 80.1   | 68.6 | 78.7 |
>
>
> [1] Saha, Swarnadeep, et al. "Learning to plan and reason for evaluation with thinking-llm-as-a-judge." arXiv preprint arXiv:2501.18099 (2025).
>
> [2] Yu, Jiachen, et al. "Improve llm-as-a-judge ability as a general ability." arXiv preprint arXiv:2502.11689 (2025).
>
> [3] Chen, Nuo, et al. "Judgelrm: Large reasoning models as a judge." arXiv preprint arXiv:2504.00050 (2025).
>
> [4] Whitehouse, Chenxi, et al. "J1: Incentivizing thinking in llm-as-a-judge via reinforcement learning." arXiv preprint arXiv:2505.10320 (2025).
>
> [5] Huang, Hui, et al. "Think-j: Learning to think for generative llm-as-a-judge." arXiv preprint arXiv:2505.14268 (2025).
>
> ---
>
> **Q4. How sensitive is RIL to the quality or diversity of teacher outputs?**
>
> Both the diversity and the quality of teacher outputs are critical to RIL’s success. As the ablation study below shows, RIL’s performance degrades when teacher responses lack diversity or accuracy. (We assume diversity is satisfied by generating a larger number of responses, since we set the inference hyperparameters—temperature, top‑p, and top‑k.)
>
> The first table reports performance as we vary the number of pre‑generated teacher responses used to train the discriminator and drive the RIL loop:
>
> | Number of Generated Responses | MathVista | MMB  | MM-Vet | MMMU | Avg  |
> |-------------------------------|:-----------:|:------:|:--------:|:------:|:------:|
> | 0 (Dr.GRPO)                   | 69.5      | 84.3 | 73.5   | 57.2 | 71.1 |
> | 2                             | 73.0      | 85.5 | 75.0   | 60.5 | 73.5 |
> | 4                             | 76.5      | 86.0 | 78.0   | 64.3 | 76.2 |
> | 8                             | 77.0      | 86.2 | 79.0   | 68.6 | 77.7 |
> | 16                            | 79.7      | 86.3 | 80.4   | 65.7 | 78.0 |
> | 32                            | 80.0      | 86.5 | 80.3   | 65.2 | 78.0 |
>
> Note that, this result is based on Qwen2.5-VL-RIL-7B (Both).
>
> We also evaluated the effect of introducing incorrect answers by randomly shuffling a percentage of generated responses relative to their questions (a dash “–” indicates mode collapse):
>
> | Shuffle Percent  | MathVista | MMB  | MM-Vet | MMMU | Avg  |
> |------------------|:-----------:|:------:|:--------:|:------:|:------:|
> | 0\%              | 79.7      | 86.3 | 80.4   | 65.7 | 78.0 |
> | 10\%             | 75.0      | 82.0 | 77.0   | 59.6 | 73.4 |
> | 30\%             | 62.0      | 68.0 | 60.0   | 30.0 | 55.0 |
> | 50\%             | -         | -    | -      | -    | -    |
> | 100\%            | -         | -    | -      | -    | -    |
>
> Note that, this result is based on Qwen2.5-VL-RIL-7B (Both).
>
> In summary, RIL benefits substantially from both diverse and accurate teacher outputs. Performance improves steadily as the number of high‑quality, varied responses increases. Conversely, introducing inconsistent or conflicting answers causes a marked degradation in performance.
>
> ---
>
> **Q5. Are there any observed failure modes during RIL training? If so, how are they mitigated?**
>
> Based on our answers to Q3 and Q4 above, we have addressed these failure modes. Briefly, to prevent them: (1) we use a structured prompt with tags and explicit reasoning for LLM-as-a-Judge, and (2) we convert the continuous reward signal into a binary one to stabilize training. Our response to Reviewer Hg1D’s Q4 provides additional details on training stabilization in the discriminator and the similarity reward signal. Furthermore, to avoid failure modes, we have already conducted the hyperparameter tuning (e.g., the number of updating parameter of discriminator and small VLMs in one training iteration, and coefficient of KL divergence), as shown in Table 5(a) and (b).
>
> ---
>
> **Q6. How is the answer parsed from the responses generated by RIL-finetuned models?**
>
> All evaluation benchmarks for VLMs already provide guidelines on how to parse responses and evaluate accuracy or other metrics. We follow each benchmark's parsing rules, in which we typically use pre-defined input prompts such as: "Answer the question using a single word or phrase", "Please answer yes or no", and "Answer with the option’s letter from the given choices directly".
>
> ---
>
> **Q7. What prompt format is used for the teacher models? For questions with deterministic answers, the similarity and answer rewards may become highly correlated, potentially limiting the extent of knowledge transfer from teacher to student.**
>
> As you mentioned, deterministic answers can clearly limit the extent of knowledge transfer from teacher to student. To address this, we sampled 40K examples from the 4M SFT dataset based on answer length, specifically to avoid deterministic or overly simple questions and instead focus on open-ended or advanced ones. We assumed that if a ground-truth answer required more than 256 tokens, it was more likely to represent an advanced problem. Additionally, we filtered for diverse domains—including image description, advanced chart-document reasoning, knowledge understanding, and multi-step or multi-dimensional solutions—to ensure a broad range of complex tasks. In this way, we made a deliberate effort to avoid deterministic questions and promote richer knowledge transfer. We will definitely add this explanation about how we constructed RIL dataset to the manuscript.
>
> ---
>
> We sincerely hope that our rebuttals have resolved your concerns and will lead to a reconsideration of the scores.

---

> > ### Comment · Reviewer_SQq6 · 2025-08-03
> > **The response solve most of my concerns**
> >
> > I appreciate the authors' thorough response to my concerns. Most of my issues have been addressed after the authors included additional sensitive experiments. As a result, I will raise my score to borderline accept.
> >
> > That said, the paper still requires major revisions, particularly in the areas of computational demands and case study. I recommend that the authors revise these aspects if the paper is accepted. However, rejecting the paper and reevaluating it through a new submission would also be acceptable.

---

> > > ### Author Response · Authors · 2025-08-03
> > >
> > > We sincerely appreciate the reviewer’s thoughtful feedback and are glad that our additional experiments helped clarify the original concerns and led to an updated score. In the revised paper, we will definitely address the computational demands with a detailed analysis of time and memory efficiency, and expand the case study to cover training stability and model collapse.

---

### Official Review · Reviewer_Hg1D · 2025-07-03

**Clarity:** 2
**Significance:** 3
**Originality:** 3
**Rating:** 5
**Confidence:** 3

**Summary:**

- This paper introduces RIL, a training framework that enables smaller vision-language models to mimic the text generation capabilities of significantly larger teacher models by combining reinforcement learning with adversarial imitation learning.
- RIL employs a dual reward system consisting of similarity rewards from an LLM-based discriminator that distinguishes between student and teacher outputs, and accuracy rewards from LLM-as-a-Judge evaluation.
- Extensive experiments across a wide array of vision-language benchmarks using multiple model architectures (Qwen2.5-VL and InternVL3 variants from 1B to 8B parameters) demonstrate consistent performance improvements, with RIL-trained models achieving competitive results with state-of-the-art open and closed-source VLMs.

**Questions:**

Please refer to the weaknesses section for details regarding the discriminator design and presentation issues.

**Ethical Concerns:**

["NO or VERY MINOR ethics concerns only"]

**Final Justification:**

I have carefully reviewed the other reviewers' comments and the authors' rebuttal.

Additional clarifications were helpful in resolving my concerns (e.g., on training details and performance analysis), and I believe most of the weaknesses I identified have been adequately addressed through extensive experiments, including the ones provided in response to the other reviewers' comments. I will maintain my original rating of 5.

**Limitations:**

Technical limitations are discussed, but potential negative societal impact is not discussed (but this doesn't appear to be critical from my understanding).

**Quality:**

3

**Strengths And Weaknesses:**

## Strengths
- The framework demonstrates impressive performance gains, with 7B parameter models achieving competitive or superior results compared to models 10x larger and proprietary systems across well-established vision-language benchmarks. Visualizations in Figure 1 is effective as well.
- The integration of GRPO with GAIL provides a principled approach to combining reinforcement learning with imitation learning for comprehensive training optimization.
- Evaluation across two distinct model families (InternVL3 and Qwen2.5-VL) with multiple parameter scales (1B to 8B) demonstrates the method's broad applicability and generalizability across different vision-language architectures.
- Comprehensive ablation studies systematically investigate major design choices.

## Weaknesses
- The framework lacks clarity in several key aspects. The initialization and training dynamics of LLM-as-a-Judge are not well explained - it's unclear whether it's trained simultaneously with SFT or how its learning evolves during RIL. Additionally, training duration and stopping conditions are not specified, and the paper doesn't adequately explain why RIL produces considerably different improvements across benchmarks.
- Converting continuous discriminator scores to binary values may discard informative signals, e.g., to which extent the responses from the student model is indiscernible. Concrete justification beyond "drawing inspiration from prior works" could help strengthen the motivation behind this design.
- There seems to be presentation issues with experimental results appearing before the proposed framework is explained (e.g., Table 1 on page 5), and excessive tables with extensive numerical results throughout the main paper. The core contribution could be communicated more effectively by moving comprehensive numerical results to the appendix while retaining only the most critical comparisons in the main text.

---

> ### Author Rebuttal · Authors · 2025-07-26
>
> **Q1. The framework lacks clarity in several key aspects. The initialization and training dynamics of LLM-as-a-Judge are not well explained - it's unclear whether it's trained simultaneously with SFT or how its learning evolves during RIL.**
>
> We would like to clarify that, as mentioned in line 164, Qwen2.5-32B is used as LLM-as-a-Judge. It is not trained during SFT or RIL steps. This model serves solely to determine whether the predicted response accurately reflects the meaning of the ground truth. We will revise the manuscript to clarify this point and avoid reader confusion.
>
> ---
>
> **Q2. Additionally, training duration and stopping conditions are not specified.**
>
> Using 256 NVIDIA A100 GPUs, pre-training the discriminator on 1.2M samples (40K [number of samples] $\times$ 16 [generated responses] $\times$ 2 [for both teacher and student]) takes approximately 1 to 3 days. The SFT step on the 4M-sample SFT dataset takes around 3 to 5 days. Conducting the RIL loop for the sampled 40K data requires an additional 3 to 5 days using 8 NVIDIA A100 GPUs. We do not apply specific stopping conditions; instead, we train one epoch but conduct four repetitive sampling per question-answer pair.
>
> ---
>
> **Q3. The paper doesn't adequately explain why RIL produces considerably different improvements across benchmarks.**
>
> We believe the performance differences of RIL across benchmarks stem from the nature of the tasks themselves. Some benchmarks require only short or factual answers, for example, "what color of t-shirt is a person wearing in the image?", (e.g, SEED2+ and RWQA), while others demand longer reasoning (e.g., MathVista, MMMU), which covers image description, advanced chart-document reasoning, knowledge understanding, and multi-dimensional or step-by-step solutions. Since reinforcement learning introduces a generalization effect, especially when mimicking the text style of large VLMs, the smaller models naturally learn to produce more coherent, logical, and well-structured responses. This is particularly beneficial in tasks that require reasoning or explanation. Therefore, RIL tends to yield greater improvements on benchmarks that benefit from such reasoning ability.
>
> | Models                   | SEED2+ | RWQA | MathVista | MMMU |
> |--------------------------|:------:|:----:|:---------:|:----:|
> | Qwen2.5-VL-7B                   |  70.4  | 68.5 |    67.8   | 55.0 |
> | Qwen2.5-VL-RIL-7B (Both) |  72.8  | 72.8 |    79.7   | 65.7 |
> | $+\Delta$ (\%)                     |   2.4  |  4.3 |    11.9    | 10.7 |
>
> ---
>
> **Q4. Converting continuous discriminator scores to binary values may discard informative signals, e.g., to which extent the responses from the student model is indiscernible.**
>
> We would like to clarify that using binary rewards enables efficient and stable training by controlling the reward steps applied. As shown in the table, a simpler reward design appears to positively impact training stability and efficiency.
>
> | Models        | Reward Step (Similarity Reward)                   | MathVista | MMB  | MM-Vet | MMMU | Avg  |
> |---------------|---------------------------------------------------|:-----------:|:------:|:--------:|:------:|:------:|
> | Qwen2.5-VL-7B | 0~1 (Continuous)                                  | 74.0      | 80.6 | 74.7   | 60.0 | 72.3 |
> | Qwen2.5-VL-7B | 0, 0.1, 0.2, 0.3, 0.4, 0.5, 0.6, 0.7, 0.8, 0.9, 1 | 75.3      | 81.9 | 76.0   | 61.3 | 73.6 |
> | Qwen2.5-VL-7B | 0, 0.5, 1                                         | 78.8      | 85.4 | 79.5   | 64.8 | 77.1 |
> | Qwen2.5-VL-7B | 0,1 (Binary, **Ours**)                            | 79.7      | 86.3 | 80.4   | 65.7 | 78.0 |
> | InternVL3-8B  | 0~1 (Continuous)                                  | 74.8      | 85.1 | 77.1   | 65.6 | 75.7 |
> | InternVL3-8B  | 0, 0.1, 0.2, 0.3, 0.4, 0.5, 0.6, 0.7, 0.8, 0.9, 1 | 76.0      | 86.3 | 78.3   | 66.8 | 76.9 |
> | InternVL3-8B  | 0, 0.5, 1                                         | 77.1      | 87.4 | 79.4   | 67.9 | 78.0 |
> | InternVL3-8B  | 0,1 (Binary, **Ours**)                            | 77.8      | 88.1 | 80.1   | 68.6 | 78.7 |
>
> There are two possible reasons for this: (a) it is difficult for models to understand why a score of 0.21 is better than 0.2, and (b) discriminator scores might be overconfident or underconfident—a well-known phenomenon in deep learning. This has led to many studies highlighting the importance of calibrated confidence. However, utilizing continuous rewards is not the focus of our work, and therefore, we do not incorporate any calibration techniques.
>
> ---
>
> **Q5. Concrete justification beyond "drawing inspiration from prior works" could help strengthen the motivation behind this design.**
>
> Prior works [1-2] have argued that natural language response-based distillation is more effective than high-dimensional feature distillation, emphasizing the importance of utilizing the language head—referred to as the verbalization effect. Motivated by this insight, we began by leveraging the similarity in natural language responses between teacher and student models through reinforcement learning and imitation learning. We will incorporate this explanation into the manuscript to better clarify the motivation behind our design.
>
> [1] Lee, Byung-Kwan, et al. "VLsI: Verbalized Layers-to-Interactions from Large to Small Vision Language Models." Proceedings of the Computer Vision and Pattern Recognition Conference. 2025.
>
> [2] Li, Yuhui, et al. "Eagle: Speculative sampling requires rethinking feature uncertainty." arXiv preprint arXiv:2401.15077 (2024).
>
> ---
>
> **Q6. There seems to be presentation issues with experimental results appearing before the proposed framework is explained (e.g., Table 1 on page 5), and excessive tables with extensive numerical results throughout the main paper. The core contribution could be communicated more effectively by moving comprehensive numerical results to the appendix while retaining only the most critical comparisons in the main text.**
>
> We will revise the structure of the paper to ensure that the proposed framework and its components are clearly introduced before presenting the corresponding experimental results, including Table 1. Additionally, we will relocate the more exhaustive numerical results to the appendix and retain only the most essential comparisons to highlight our core contributions.
>
> ---
>
> Thank you for your encouraging review. We hope our responses have further clarified the points you raised.

---

> > ### Comment · Reviewer_Hg1D · 2025-08-04
> >
> > I have carefully reviewed the other reviewers' comments and the authors' rebuttal.
> >
> > Additional clarifications were helpful in resolving my concerns, and I believe most of the weaknesses I identified have been adequately addressed through extensive experiments. I will maintain my original rating of 5.

---

> > > ### Author Response · Authors · 2025-08-04
> > >
> > > Thank you very much for your thorough review and for maintaining your score. We truly appreciate your thoughtful feedback and engagement throughout the process.

---

### Decision · Program_Chairs · 2025-09-17

**Decision:**

Accept (poster)

**Comment:**

This paper introduces a new approach to training small VLMs via mimicking the behaviour of larger ones based on a newly proposed Reinforcement Learning with Imitation Learning framework. All reviewers agree on the novelty of the proposed, and that the paper presents some strong results on an impactful research problem. Almost all weaknesses were addressed during rebuttal and authors also promised to address issues raised by SQq6 regarding the computational demands with a detailed analysis of time and memory efficiency, Hence, clear accept.